# The levels, variation characteristics and sources of atmospheric non-methane hydrocarbon compounds during wintertime in Beijing, China

Chengtang Liu[1,3 ⊥], Zhuobiao Ma[1,3 ⊥], Yujing Mu[1,2,3,4] *, Junfeng Liu[1,3], Chenglong Zhang[1,3], Yuanyuan Zhang[1,3], Pengfei Liu[1,3], Hongxing Zhang[1,5]

[1]Research Center for Eco-Environmental Sciences, Chinese Academy of Sciences, Beijing 100085, China
[2]Center for Excellence in Regional Atmospheric Environment, Institute of Urban Environment, Chinese Academy of Sciences, Xiamen 361021, China
[3]University of Chinese Academy of Sciences, Beijing 100085, China
[4]National Engineering Laboratory for VOCs Pollution Control Material & Technology, University of Chinese Academy of Sciences, Beijing 100049, China
[5]Beijing Urban Ecosystem Research Station, Beijing, 100085, China

*Correspondence to*: Yujing Mu (yjmu@rcees.ac.cn)

**Abstract.** Atmospheric non-methane hydrocarbon compounds (NMHCs) were measured at a sampling site in Beijing city from 15 December 2015 to 14 January 2016 to recognize their pollution levels, variation characteristics and sources. Fifty-three NMHCs were quantified and the proportions of alkanes, alkenes, acetylene and aromatics to the total NMHCs were 49.8% ~ 55.8%, 21.5% ~ 24.7%, 13.5% ~ 15.9% and 9.3% ~ 10.7%, respectively. The variation trends of the NMHCs concentrations were basically identical and exhibited remarkable fluctuation, which were mainly ascribed to the variation of meteorological conditions, especially wind speed. The diurnal variations of NMHCs in clear days exhibited two peaks during the morning and evening rush hours, whereas the rush hours' peaks diminished or even disappeared in the haze days, implying that the relative contribution of the vehicular emission to atmospheric NMHCs depended on the pollution status. Two evident peaks of the propane/propene ratios respectively appeared in the early morning before sun rise and at noontime in clear days, whereas only one peak occurred in the afternoon during the haze days, which were attributed to the relatively fast reactions of propene with OH, $NO_3$ and $O_3$. Based on the chemical kinetic equations, the daytime OH concentrations were calculated to be in the range of $3.47 \times 10^5$ - $1.04 \times 10^6$ molecules·cm$^{-3}$ in clear days and $6.42 \times 10^5$ - $2.35 \times 10^6$ molecules·cm$^{-3}$ in haze days, and the nighttime $NO_3$ concentrations were calculated to be in the range of $2.82 \times 10^9$ - $4.86 \times 10^9$ molecules·cm$^{-3}$ in clear days. The correlation coefficients of typical hydrocarbons pairs (benzene/toluene, o-xylene/m,p-xylene, isopentane/n-pentane, etc.) revealed that vehicular emission and coal combustion were important sources for atmospheric NMHCs in Beijing during the wintertime. Five major emission sources for atmospheric NMHCs in Beijing during the wintertime were further identified by positive matrix factorization (PMF), including gasoline related emissions (gasoline exhaust and evaporation), coal combustion, diesel exhaust, acetylene-related emission and consumer and household products. Coal combustion (probably domestic coal combustion) were found to make the greatest contribution (29.6~33.4%) to atmospheric NMHCs during haze days.

# 1 Introduction

As an important class of volatile organic compounds (VOCs), non-methane hydrocarbons (NMHCs) play pivotal role in atmospheric chemistry (Houweling et al., 1998; Rappengluck et al., 2014) and their degradation can cause formation of secondary products (such as ozone ($O_3$) and secondary organic aerosols (SOA)) which affect the oxidizing capacity, radioactive balance, and human health (Volkamer et al., 2006; Shen et al., 2013; Huang et al., 2014; Liu et al., 2015; Palm et al., 2016; La et al., 2016). NMHCs can originate either from biogenic or anthropogenic sources. Biogenic sources are mainly from emission of vegetation and anthropogenic sources are related to fossil fuel combustion (vehicle exhaust, heat generation and industrial processes), storage and distribution of fuels (gasoline, natural gas and liquefied petroleum gas) and solvent use. Because the global emission and the reaction activity of biogenic NMHCs are much greater than those of anthropogenic NMHCs (Goldstein and Galbally, 2007), atmospheric biogenic NMHCs (e.g., isoprene) are more important in global atmospheric environment. In urban areas, however, anthropogenic NMHCs greatly exceed biogenic NMHCs and have been considered as one of the most dominant drivers of air pollution (Srivastava et al., 2005; Gaimoz et al., 2011; Waked et al., 2012). In addition, some anthropogenic NMHCs (e.g., benzene and 1,3-butadiene) have been verified to be toxic, carcinogenic or mutagenic (US EPA, 2008; Møller et al., 2008). Due to the negative impact of NMHCs on atmospheric environment as well as human health, atmospheric NMHCs measurements have been world widely conducted in many urban areas (Shirai et al., 2007; Gaimoz et al., 2011; Waked et al., 2016), and the results revealed that NMHCs made remarkable contribution to atmospheric $O_3$ and SOA in most cities and the cancer risk of benzene evenly exceeded the value of $1.0 \times 10^{-6}$ in some cities (Zhou et al., 2011; Du et al., 2014).

Beijing, as one of the world's megacities, has been encountering two prominent atmospheric environmental problems: the elevation of near-surface $O_3$ levels and the serious pollution of fine particles (including SOA) which result in frequent haze formation (Sun et al., 2014). Therefore, the levels and sources for atmospheric NMHCs in Beijing city have been aroused great concern (Song et al., 2007; Wu et al., 2016). More than 40 papers about NMHCs in Beijing city have been published since 1994 (Shao et al., 1994), and the results indicated that the concentrations of NMHCs in Beijing were evidently higher than those in the cities of most developed countries (Gros et al., 2007; Parrish et al., 2009). The major components of atmospheric NMHCs in Beijing city were found to be alkanes, alkenes and aromatics, and the relatively high proportions of alkenes and aromatics have been suspected to be responsible for formation of $O_3$ and SOA (Li et al., 2015; Sun et al., 2016). Based on the model of positive matrix factorization (PMF), several studies also investigated the major sources and their contributions to atmospheric NMHCs in Beijing city: transportation-related sources (32~46%), paint and solvent use and industry (18~30%), solvent utilization (8~14%) were found to be the major sources for atmospheric NMHCs in summer (Wang et al., 2015; Li et al., 2016), while vehicle exhaust (26~39%) and coal combustion (35~41%) were the dominant sources in winter (Wang et al., 2013). However, most studies mainly focused on summertime, the data of atmospheric NMHCs in Beijing city during wintertime were still sparse, e.g., only two reports about atmospheric NMHCs (Wang et al., 2012; Wang et al., 2013) and two reports about benzene, toluene, ethylbenzene and xylene (BTEX) (Zhang et al., 2012a;

Zhang et al., 2012b). To comprehensively evaluate the influence of atmospheric NMHCs on the air quality and to further identify the sources of atmospheric NMCHs in Beijing city, more measurements of atmospheric NMHCs in winter are still needed.

In this study, atmospheric NMHCs were online measured using a liquid nitrogen-free gas chromatography-flame ionization detector (GC-FID) in Beijing city from 15 December 2015 to 14 January 2016. The objectives of this study are (1) to determine the concentration levels and variation characteristics of atmospheric NMHCs in Beijing during wintertime; (2) to identify the major sources for atmospheric NMHCs in Beijing during wintertime.

## 2 Experimental

### 2.1. Sampling site description

Air samples were collected on a rooftop (20 m above the ground level) in Research Center for Eco-Environmental Sciences (RCEES, 40.0°N, 116.3°E) which lies in the north of Beijing city between the 4th and 5th rings roads. The sampling site is surrounded by some residential areas, campuses and institutes. The detail information about the sampling site was described in our previous studies (Pang and Mu, 2006; Liu et al., 2009). The meteorological data, including temperature, wind speed, relative humidity (RH), visibility and Air Pollution Index of particulate matter with diameter of less than 2.5 μm ($PM_{2.5}$) at RCEES were from Beijing urban ecosystem research station, which is about 20 m away from our sampling site.

### 2.2. Analytical methods

Air samples were analyzed continuously and automatically using a custom-built/liquid nitrogen-free GC-FID online instrument, with a time resolution of 1 h. The online instrument is mainly consisted of a cooling unit, a sampling unit, a separation unit and a detection unit, and the detailed description of the analytical instrument is described in Liu et al. (2016a). Briefly, a sample amount of 400 mL (50 mL·min$^{-1}$ × 8 min) of the air was pre-concentrated in a stainless steel tube filled with Carbopack$^{TM}$ B adsorbent (60/80 mesh). Once the pre-concentration was finished, the adsorption tube was quickly heated to about 100℃ and the NMHCs desorbed were injected into a single column (OV-1, 30 m × 0.32 mm I.D.) for separation. The temperature program of the capillary column used was as follows: 3 min at -60℃, ramped at 12℃ min$^{-1}$ to -20℃, ramped at 6℃ min$^{-1}$ to 30℃, ramped at 10℃ min$^{-1}$ to 170℃, then hold for 2 min. The detection unit is a FID, and the temperature of the FID is operated at 250℃.

We used an external standard method for the quantification of C2-C12 hydrocarbons by diluting 1.0 ppmv standard gas mixtures of 57 NMHCs (provided by Spectra Gases Inc., USA) with high pure nitrogen gas. Five concentrations (1.0-30.0 ppbv) were used to perform calibrations. $R^2$ values for calibration curves were all above 0.99 for NMHCs, indicating that integral areas of peaks were proportional to concentrations of target compounds. We performed weekly calibrations, and the variations in target species responses were within 6 % of the calibration curve. The method detection limits (MDLs) of 0.02-

0.10 ppbv for the NMHCs were estimated based on the signal to noise ratio of 3 and enrichment volume of 400 ml (Liu et al., 2016a).

## 2.3 PMF model analysis

The US PMF 5.0 was applied to identify major emission sources of NMHCs sources (Sowlat et al., 2016). PMF is a multivariate factor analysis tool that decomposes a matrix of speciated sample data into two matrices-factor contributions and factor profiles which can be interpreted by an analyst as to what sources are represented based on observations at the receptor site (Guo et al., 2010; Ling et al., 2011; Ou et al., 2015; Shao et al., 2016, Shi et al., 2009;Xie et al., 2008;Lanz et al., 2007;Zhang et al., 2013). The object function Q, based on the uncertainties inherent in each observation, can allow the analyst to review the distribution for each species to evaluate the stability of the solution:

$$Q = \sum_{i=1}^{m} \sum_{j=1}^{n} \left[ \frac{x_{ij} - \sum_{k=1}^{p} g_{ik} f_{kj}}{u_{ij}} \right]^2 \tag{1}$$

where $u_{ij}$ is the uncertainty estimate of source $j$ measured in sample $i$, $x_{ij}$ is the $j$th species concentration measured in the $i$th sample, $g_{ik}$ is the species contribution of the $k$th source to the $i$th sample, $f_{kj}$ is the $j$th species fraction from the $k$th source, and $p$ is the total number of independent sources.

For the PMF input, it is not necessary to use all of the measured NMHCs for the PMF model due to the fundamental assumption of non-reactivity and/or mass conservation of the PMF model (Guo et al., 2011a;Ling and Guo, 2014). The selection of the NMHCs species for the input of the PMF model was based on the adopted principles in previous studies (Ling and Guo, 2014). In total, 17 major NMHCs, which accounted for about 90% (ppbv/ppbv) of the total concentrations of the measured NMHCs species, were input into the PMF model to explore the sources of observed NMHCs. The uncertainty of input data is another input required by PMF. According to the method recommended by EPA PMF Fundamentals, the uncertainties for each sample/species with values high the detection limit (DL) were calculated using the following equation:

$$uncertainty = \sqrt{Precision^2 + DL^2} \tag{2}$$

where the precision accounts for the relative measurement error determined by calibration of the instruments. Values below the detection limit were replaced by 1/2 of the DL and their overall uncertainties were set at 5/6 of the DL values. In this analysis, different numbers of factors were tested. The robust mode was used to reduce the influence of extreme values on the PMF solution. In addition, many different starting seeds were tested and no multiple solutions were found. More than 95% of the residuals were between -3 and 3 for all compounds. The Q values in the robust mode were approximately equal to the degrees of freedoms. These features demonstrated that the model simulation results were acceptable.

## 3 Results and discussion

### 3.1 The levels and variation characteristics of NMHCs during the sampling period

Fifty-three NMHCs were quantified and classified into alkanes, alkenes, aromatics and acetylene. The variations of total

NMHCs (TNMHCs), alkanes, alkenes, aromatics and acetylene together with meteorological parameters during the measurement period are shown in Fig. 1. It is evident that the variation trends of the NMHCs concentrations were basically identical and exhibited significantly fluctuation, which were mainly ascribed to the variation of surface wind speed, e.g., the TNMHCs concentrations were lower than 30 ppbv when wind speeds were greater than 2 m·s$^{-1}$, whereas sharply increased as the wind speed decreased. Although the surface wind speeds were relatively higher on 25-26 December 2015 than other days with pollution episodes, the concentrations of NMHCs were the highest. The relatively stable and low air temperature during 25-26 December 2015 (Fig. 1D) indicated that the surface wind with the cold air might result in advection inversion which favored accumulation of the pollutants. The daily average concentration of TNMHCs increased from about 30 ppbv to about 100 ppbv within 3-6 days during the three pollution episodes on 17-22 December, 27-29 December and 31 December-3 January, whereas increased from about 30 ppbv to 165 ppbv within one day during the pollution episode on 25-26 December. The strong wind only lasted 5 h before the sever pollution episode on 25-26 December, whereas the strong wind events lasted at least one day before the other pollution episodes. High concentrations of the pollutants after the strong wind event with duration of short period on 24 December were suspected to distribute in the neighbor of Beijing, which could accelerate the accumulation of the pollutants in Beijing after the strong wind event.

It should be noted that the variation trend of TNMHCs was almost same as that of PM$_{2.5}$, and significant linear correlation with coefficient ($R^2$) of 0.9 was found. In contrast to NMHCs and PM$_{2.5}$, ozone concentrations approached to zero during each haze events and reached to the maximum of about 35 ppbv in daytime just after the haze events followed by strong winds from northwest directions (Lin et al., 2011). Although strong winds from northwest directions occurred during the period of 12-14 January 2016, ozone concentrations didn't evidently increase during daytime, implying that ozone formation depended on the pollution levels of its precursors (NMHCs and NOx). Because NMHCs are solely from direct emissions and PM$_{2.5}$ is from both direct emissions and secondary formation, the almost same variation trends of them further indicated that meteorological conditions, especially the surface wind speed played pivotal role for their accumulation and dispersion. On the other hand, some of the NMHCs (e.g., aromatics) measured are the precursors for SOA, the remarkable elevation of the NMHCs during pollution episodes would also make more contribution to PM$_{2.5}$ through SOA formation because of relatively high OH radical concentration during the pollution episodes (see Sect. 3.2.2). It should be also mentioned that odd-even license plate number rule was adopted in Beijing on 19-22 December 2015. Compared with the two pollution events on 27-29 December and 31 December-3 January, although the wind speeds were slightly faster, the peak values and the daily average concentrations of TNMHCs and PM$_{2.5}$ during the period of 17-22 December 2015 were almost same as the two pollution events without adopting the rule, implying that the sources other than vehicle emission might be dominant for atmospheric NMHCs and PM$_{2.5}$ during the haze days.

The entire sampling period was divided into three categories based on the daily average visibility values: clear days ($\geq$ 10 km, RH < 90%), light haze days (5-10 km, RH < 90%) and heavy haze days ($\leq$ 5 km, RH < 90%) (Yang et al., 2012; Lin et al., 2014). As shown in Table 1, there were 11 heavy haze days, 8 light haze days and 12 clear days during the 31 sampling days (non-precipitation days). The mean concentrations and standard deviations of TNMHCs, alkanes, alkenes, aromatics

and acetylene during the three categories are presented in Table 2. It is evident that the average concentrations of alkanes, alkenes, aromatics, acetylene and TNMHCs remarkably increased from clear days to heavy haze days, and their average concentrations during heavy haze days were at least a factor of 5 higher than those during clear days, which were in good agreement with the previous studies (Zhang et al., 2014). Alkanes accounted for the largest proportions (49.8% ~ 55.8%), followed by alkenes (21.5% ~ 24.7%), acetylene (13.5% ~ 15.9%) and aromatics (9.3% ~ 10.7%). The proportion order of alkenes, acetylene and aromatics obtained by this study in winter were different from that reported by previous studies in summer (Wang et al., 2010; Li et al., 2016) which was probably due to the different sources (e.g., additional domestic coal combustion in winter) for atmospheric NMHCs in Beijing between the two seasons. The top ten NMHCs measured in this study are presented in Table 3 for comparison with those from the cities in China. The average concentrations of the top ten NMHCs observed in this study were basically within the values reported in various Chinese cities (Barletta et al., 2005; Song et al., 2012; An et al., 2014; Li et al., 2015; Zou et al., 2015; Jia et al., 2016). During haze days, the average concentrations of the NMHCs in winter of Beijing were less than those reported in Foshan (Guo et al., 2011b), whereas remarkably greater than those reported in summer of Beijing (Guo et al., 2012). With only exception of isobutane and isopentane, the average concentrations of other eight NMHCs during both the whole measurement period and haze days in this study were evidently greater than those reported in summer of Beijing. The relatively higher NMHCs measured in this study during the wintertime than those measured in summer were suspected to be due to the different meteorological conditions and different sources between the two seasons.

## 3.2 Diurnal variations of NMHCs and the propane/propene ratios

### 3.2.1 Diurnal variations of NMHCs

Diurnal variations of alkanes, alkenes, aromatics and acetylene under different visibility levels are presented in Fig. 2. The two obvious peaks for the NMHCs in the morning and evening rush hours under clear days (Fig. 2a) indicated that exhaust of vehicles was an important source for atmospheric NMHCs in Beijing. For light haze days (Fig. 2b), only small peak of NMHCs could be observed during morning rush hours, and the concentration of NMHCs steady increased from 18:00 to 01:00 of the next day. For heavy haze days (Fig. 2c), the peak levels of NMHCs during the two rush hours disappeared, and the concentration of NMHCs steady increased from 17:00 to 20:00 and began to level off until 07:00 of the next day. The three distinct diurnal variations of NMHCs under the three typical days were suspected to relate with the diurnal variations of boundary layer. Besides the highest PBL height during clear days (Zheng et al., 2015;Lin et al., 2011;Zhang et al., 2014), the highest wind speed during both nighttime and daytime in clear days also favored diffusion of pollutants, resulting in the lowest levels of atmospheric NMHCs in clear days. Both the relatively high boundary layer (Gao et al., 2015; Quan et al., 2013; Liu et al., 2013) and wind speeds could result in the lowest levels of the pollutants during nighttime, which were suspected to make the peak levels of atmospheric NMHCs more evident during daytime rush hours.

### 3.2.2 Diurnal variations of propane/propene ratios

Considering the large difference of the reactivity between propane and propene towards $NO_3$ and OH radicals, the diurnal variations of propane/propene ratios were analyzed for revealing the nighttime and daytime reaction processes. Diurnal variations of propane/propene ratios under clear days, light haze days and heavy haze days are shown in Fig. 3. Two evident peaks of the propane/propene ratios respectively appeared in the early morning (about 5:00) before sun rise and at noontime (about 12:00) in clear days, whereas only one peak occurred around 15:00 during the haze days. Distinct peaks of the propane/propene ratios were mainly ascribed to the different reactivity of propane and propene, because of their possible common sources indicated by the significant correlation ($R^2=0.8$) between propane and propene during the measurement period. The atmospheric reactions of propane and propene include:

$$\text{Propane} + \text{OH} \xrightarrow{k_1} \text{Products} \qquad k_1 = 1.09 \times 10^{-12} \text{ cm}^3\cdot\text{molecule}^{-1}\cdot\text{s}^{-1} \text{ (Atkinson, 2003)} \tag{3}$$

$$\text{Propene} + \text{OH} \xrightarrow{k_2} \text{Products} \qquad k_2 = 2.57 \times 10^{-11} \text{ cm}^3\cdot\text{molecule}^{-1}\cdot\text{s}^{-1} \text{ (Daranlot et al., 2010)} \tag{4}$$

$$\text{Propene} + \text{O}_3 \xrightarrow{k_3} \text{Products} \qquad k_3 = 1.06 \times 10^{-17} \text{ cm}^3\cdot\text{molecule}^{-1}\cdot\text{s}^{-1} \text{ (Wegener et al., 2007)} \tag{5}$$

$$\text{Propane} + \text{NO}_3 \xrightarrow{k_4} \text{Products} \qquad k_4 = 7.00 \times 10^{-17} \text{ cm}^3\cdot\text{molecule}^{-1}\cdot\text{s}^{-1} \text{ (Atkinson et al., 2001)} \tag{6}$$

$$\text{Propene} + \text{NO}_3 \xrightarrow{k_5} \text{Products} \qquad k_5 = 9.54 \times 10^{-15} \text{ cm}^3\cdot\text{molecule}^{-1}\cdot\text{s}^{-1} \text{ (Atkinson et al., 2001)} \tag{7}$$

$$\text{O}_3 + \text{NO}_2 \xrightarrow{k_6} \text{NO}_3 + \text{O}_2 \qquad k_6 = 3.52 \times 10^{-17} \text{ cm}^3\cdot\text{molecule}^{-1}\cdot\text{s}^{-1} \text{ (Atkinson et al., 2004)} \tag{8}$$

The rate constants for the reactions of OH and $NO_3$ with propene are a factor of 2.4 and 136.3 greater than with propane, respectively. In addition, $O_3$ can react with propene but not with propane. The evident $O_3$ concentrations (about 25 ppbv) during nighttime in clear days could react with $NO_2$ to form $NO_3$ radicals via reaction (8), whereas the formation of $NO_3$ radicals was completely blocked during haze days because $O_3$ concentrations were extremely low (nearly zero). Therefore, the peak of the propane/propene ratios appeared in nighttime during clear days was rationally ascribed to the additional consumption of propene by $O_3$ and $NO_3$. Based on the data measured, the OH and $NO_3$ concentrations were roughly estimated according to following chemical kinetic equations (The Eq. (9) - (12) were used to estimate the concentrations of OH during daytime and Eq. (13) - (16) were used to estimate the concentrations of $NO_3$ during nighttime):

$$[\text{Propane}]_0 = [\text{Propane}]_t \times e^{k_1[\text{OH}]\Delta t} \tag{9}$$

$$[\text{Propene}]_0 = [\text{Propene}]_t \times e^{(k_2[\text{OH}]+k_3[\text{O}_3])\Delta t} \tag{10}$$

$$\ln\frac{[\text{Propane}]_t}{[\text{Propene}]_t} = \{(k_2 - k_1)[\text{OH}] + k_3[\text{O}_3]\}\Delta t + \ln\frac{[\text{Propane}]_0}{[\text{Propene}]_0} \tag{11}$$

$$[\text{OH}] = \frac{1}{k_2-k_1} \times \left\{\left(\ln\frac{[\text{Propane}]_t}{[\text{Propene}]_t} - \ln\frac{[\text{Propane}]_0}{[\text{Propene}]_0}\right)/\Delta t - k_3[\text{O}_3]\right\} \tag{12}$$

$$[\text{Propane}]_0 = [\text{Propane}]_t \times e^{(k_4[\text{NO}_3])\Delta t} \tag{13}$$

$$[\text{Propene}]_0 = [\text{Propene}]_t \times e^{(k_5[\text{NO}_3]+k_3[\text{O}_3])\Delta t} \tag{14}$$

$$\ln\frac{[\text{Propane}]_t}{[\text{Propene}]_t} = \{(k_5 - k_4)[\text{NO}_3] + k_3[\text{O}_3]\}\Delta t + \ln\frac{[\text{Propane}]_0}{[\text{Propene}]_0} \tag{15}$$

$$[NO_3] = \frac{1}{k_5 - k_4} \times \left\{ \left( \ln \frac{[Propane]_t}{[Propene]_t} - \ln \frac{[Propane]_0}{[Propene]_0} \right) / \Delta t - k_3[O_3] \right\} \qquad (16)$$

Here, [OH] is the average OH radical concentration (molecules·cm$^{-3}$), [NO$_3$] is the average NO$_3$ radical concentration (molecules·cm$^{-3}$), [O$_3$] is the average ozone concentration (molecules·cm$^{-3}$), $\Delta$t is the exposure time (s) of OH or NO$_3$, [Propane]$_0$ and [Propene]$_0$ are their initial concentrations when the propane/propene ratio began increase, [Propane]$_t$ and [Propene]$_t$ are their concentrations at $t$ (s) during the period of increasing propane/propene ratios.

Good linear correlations (R$^2$≥ 0.9) between ln $\frac{[Propane]_t}{[Propene]_t}$ and $t$ were found during the period from 9:00 to 14:00 for most days and during the period of about 0:00-5:00 for most clear days, indicating that the concentration variations of propane and propene basically abided by the above chemical kinetic rules. The OH concentrations were calculated to be in the range 3.47 × 10$^5$ - 1.04 × 10$^6$ molecules·cm$^{-3}$ in clear days and 6.42 × 10$^5$ - 2.35 × 10$^6$ molecules·cm$^{-3}$ in haze days. The relatively high OH concentrations during haze days in winter of Beijing could accelerate oxidation of gas species and further promoted formation of secondary particles. The NO$_3$ concentrations were calculated to be in the range from 2.82 × 10$^9$ molecules·cm$^{-3}$ to 4.86 × 10$^9$ molecules·cm$^{-3}$ in clear days, which were in good agreement with the maximal value (4.92 × 10$^9$ molecules·cm$^{-3}$) reported in the Houston city during winter (Asaf et al., 2010). It should be mentioned that the OH and NO$_3$ derived from the propane/propene ratios could only represent their lower limits because of the continue mixing of fresh emissions with the aged air.

### 3.3 Sources of NMHCs

### 3.3.1 The indicator of typical ratios

The ratios of o-xylene/m,p-xylene and cis-2-butene/trans-2-butene have been widely used as the indicators for gasoline vehicle exhaust emissions (Velasco et al., 2007; Li et al., 2015). As shown in Fig. 4 A-B, the slopes (0.36, 0.94) of the linear regressions between the two hydrocarbons pairs of o-xylene/m,p-xylene and trans-2-butene/cis-2-butene were in good agreement with the ratios (0.35, 1.14) from vehicle emissions (Liu et al., 2008; Wang et al., 2010), implying that vehicle emissions were their dominant source in winter of Beijing. The ratios of propane/n-butane and propane/isobutane have been frequently used for distinguishing the contributions of gasoline vehicles and the vehicles fueled with (LPG) (Liu et al., 2008; Lai et al., 2009). The slopes of propane/n-butane (3.12) and propane/isobutane (5.98) both fell between the emission ratios of gasoline vehicles (0.49 and 0.74, respectively) and vehicles using LPG (6.12 and 9.12, respectively), suggesting that emissions from both gasoline and LPG vehicles might be their important sources in Beijing. However, the ratios of propane/n-butane (1.65 - 1.94) and propane/isobutane (1.52 - 1.97) reported in summer of Beijing (Wang et al., 2010) were much less than the values obtained in this study during wintertime in Beijing. Considering the relatively stable proportion of LPG vehicles to gasoline vehicles during the whole year, additional sources were suspected to make evident contribution to the relatively high ratios of propane/n-butane and propane/isobutane in winter of Beijing. It should be mentioned that the ratios of propane/n-butane and propane/isobutane in winter of Beijing are close to those from domestic coal combustion

(4.34 and 8.68, respectively) (Liu et al., 2016b), and the ratio (1.92) of isobutane/n-butane was coincident with that from domestic coal combustion (2.0). Therefore, domestic coal combustion around Beijing in winter might make remarkable contribution to the C3-C4 alkanes. The contribution of domestic coal combustion to atmospheric NMHCs in winter of Beijing could also be confirmed by other ratios of hydrocarbon pairs. As shown in Fig. 5, the ratios of isopenatne/n-pentane, propane/isopentane, benzene/toluene and benzene/ethylbenzene were all fell between the emission ratios of vehicles and coal combustion.

Besides the above hydrocarbon pairs, the slopes of another hydrocarbon pairs were also analyzed and listed in Table 4. With exception for the hydrocarbon pairs of propane/toluene, propane/isopentane, propane/n-butane and propane/isobutane, the slopes of other pairs were within the values reported in different cities (Liu et al., 2008; Louie et al., 2013). The slopes for the hydrocarbon pairs of propane/toluene, propane/isopentane, propane/n-butane and propane/isobutane were remarkably greater than those reported in various cities including Beijing, which were suspected to be from the contribution of domestic coal combustion in winter around Beijing (see above discussion). Barletta et al. (2005) found that the slopes of benzene/acetylene, ethylene/acetylene and benzene/ethylbenzene in 15 Chinese cities with B/T >1 were remarkably greater than those in 10 Chinese cities (traffic related cities) with B/T of about 0.6, and attributed the relatively high slopes in the 15 Chinese cities to the emissions from biofuel and charcoal combustion. The slopes of benzene/acetylene, ethylene/acetylene and benzene/ethylbenzene obtained by this study were coincident with those in the 15 cities reported by Barletta et al. (2005), indicating that domestic coal combustion in winter around Beijing might make contribution to the species. It is interesting to be noted that the slopes or the ratios of o-xylene/m,p-xylene and trans-2-butene/cis-2-butene which have high OH radical reactivity in various cities were in good agreement with those of vehicle emissions, whereas the slopes or ratios of the hydrocarbon pairs with low OH reactivity showed obvious difference among the cities, implying that the atmospheric NMHCs with high OH reactivity are dominated by local emissions and the atmospheric NMHCs with low OH reactivity are strongly influenced by regional transportation.

### 3.3.2 The source profiles and apportionments of NMHCs

The PMF model was performed based on the 740 samples collected and the NMHCs species with high uncertainty were excluded to reduce the possible bias of the modeling results. Eventually, 17 NMHCs species were selected for the source apportionment analysis since they are the most abundant species and/or are typical tracers of various emission sources. The sources' appointments of atmospheric NMHCs at the receptor site for the clear days, light haze days, heavy haze days and the whole days were separately analysed by the PMF model, and similar source profiles were found. As shown in Fig. 6 for the whole database, five factors were resolved from running the PMF model designated as source 1, source 2, source 3, source 4 and source 5.

Source 1 was characterized by high percentages of iso/n-pentanes, aromatics and other C2-C7 alkanes. NMHCs from vehicular emission have been found to be dominated by iso/n-pentanes and aromatics with the benzene/toluene mass ratio of about 0.6 (Barletta et al., 2005), which was in agreement with PMF results for source 1. Additionally, C3-C5 alkanes are also

emitted from gasoline evaporations, e.g., isopentane is a typical tracer for gasoline evaporation (Liu et al., 2008). Therefore, source 1 is rationally ascribed to gasoline related emissions (gasoline exhaust and evaporation).

Source 2 was associated with high percentages of acetylene, C2-C3 alkenes, C2-C5 alkanes and benzene. It is known that acetylene is a typical species from combustion process (Barletta et al., 2005; Wu et al., 2016), and high concentrations of C2-C3 alkenes, C2-C5 alkanes and benzene have been found from resident coal combustion (dos Santos et al., 2004; Liu et al., 2008; Liu et al., 2016b). In addition, the ratios of benzene/toluene and propane/isopentane obtained from coal combustion were 1.54-2.22 (Liu et al., 2008), 8.68 (Liu et al., 2016b), respectively, which were close to the ratios in the second source profile. Source 2 has, therefore, been assigned to coal combustion.

Source 3 was associated with over 60% of the total measured acetylene, and this source is designated as acetylene-related emissions. Source 4 was dominated by high content of toluene, ethylbenzene and xylenes. It is known that these species can be emitted from coal combustion, vehicular exhaust or associated with the solvent emissions of paints, inks, sealant, varnish and thinner for architecture and decoration (Borbon et al., 2002; Guo et al., 2011a). Coal combustion and gasoline exhaust could be excluded as the main contributors to source 4, because aromatics emissions from the two sources are usually accompanied by high emissions of various species with carbon numbers less than six. Solvent emissions could also be excluded due to the relatively high contribution of small molecules such as ethylene and propene in source 4. Based on the PMF analysis for the diurnal variation characters, source 4 is finally attributed to diesel exhaust.

Source 5 was characterized by high levels of n-hexane. n-Hexane is a common constituent of glues used for shoes, leather products and roofing. Additionally, it is used in solvents to extract oils for cooking and as a cleansing agent for shoe, furniture and textile (Kwon et al., 2007; Guo et al., 2011a). Therefore, this source is identified as consumer and household products.

The time series of the contributions from the five factors to atmospheric NMHCs are shown in Fig. 7. In general, the variation trends of the contributions from gasoline related emissions (gasoline exhaust and evaporation), diesel exhaust, coal combustion emissions and acetylene-related emissions to atmospheric NMHCs were closely related with the variation trend of atmospheric NMHCs measured, while the contribution from the consumer and household products had less correlation with the atmospheric NMHCs measured. The daily emissions from gasoline related sources (gasoline exhaust and evaporation), diesel exhaust, coal combustion sources and acetylene-related sources are usually stable, and hence, the similar variation trends of their contributions to atmospheric NMHCs were mainly ascribed to the variation of meteorological condition. The sources of consumer and household products were suspected to be irregular for explaining the abnormal variation trends of their contributions to atmospheric NMHCs. It should be mentioned that the contribution from coal combustion was the maximum during the most serious pollution episode II (25-26 December 2015) when the wind direction was from southwest, implying that the air parcel transportation from southern was an important source for NMHCs in Beijing (Wang et al., 2013).

The diurnal variations of the contributions from the five factors to atmospheric NMHCs are shown in Fig. 8. Compared with the sources of coal combustion, acetylene-related emissions and consumer and household products, the contributions of the

vehicle emissions (gasoline and diesel exhaust) to atmospheric NMHCs during the morning and evening rush hours indeed evidently increased during the clear days and light haze days, but slightly decreased in the morning rush hours during the heavy haze days. The remarkably higher contributions of diesel exhaust than gasoline emissions during the midnight for haze days well reflected the traffic situation, namely, heavy diesel vehicles being only permitted on the road during the midnight

in Beijing. The relatively high contributions of consumer and household products to atmospheric NMHCs mainly occurred in clear days during daytime when temperature was relatively high. No distinct diurnal variations of the contributions from coal combustion and acetylene-related emissions to atmospheric NMHCs were found.

Fig. 8 also shows the individual contributions of the five major NMHCs sources to the NMHCs concentrations measured in clear days, light haze days and heavy haze days. It is clear that the share rates of the five major sources to atmospheric

NMHCs under the three typical days varied significantly. Gasoline exhaust and evaporation was the largest contributor in clear days, followed by diesel exhaust, coal combustion, acetylene-related emission and consumer and household products, whereas coal combustion made the largest contribution in haze days, followed by Gasoline exhaust and evaporation, diesel exhaust, acetylene-related emission and consumer and household products. Considering the daily emissions of NMHCs from the five major sources were relatively stable during the short period of the winter, the distinct variation of the share rates

from the five major sources under the three typical days was suspected to be related to the reactivity of the dominant species from each source because of the evidently different OH concentrations between clear days and haze days (see Sect. 3.2.2). Compared with the species from the other four major sources, the dominant species of toluene, ethylbenzene and xylenes emitted from the diesel exhaust are highly reactive, and hence, remarkable decrease of the share rate from this source was observed from clear days to haze days. The dominant species of alkanes from coal combustion were relatively stable in

comparison with those (alkenes and aromatics) from gasoline exhaust and evaporation, resulting in the fast increase of the share rate from coal combustion from clear days to haze days. Although the central heating stoves that used coal as energy in Beijing have been replaced by the relatively clean energies, coal combustion was still an important source for ambient NMHCs during wintertime in Beijing. It should be mentioned that domestic coal combustion is prevailing for heating and cooking by farmers in rural areas around Beijing city, e.g., the domestic coal consumption account for about 11% of the total

in the region of Beijing-Tianjin-Hebei (http://www.qstheory.cn/st/dfst/201306/t20130607_238302.htm). Additionally, the emission factors of NMHCs from domestic coal combustion have been found to be a factor of 20 greater than those from coal power plants (Liu et al., 2016b). Therefore, the high share rate of coal combustion in Beijing city was mainly attributed to the regional transportation.

## 4. Conclusions

Atmospheric non-methane hydrocarbon compounds (NMHCs) were measured at a sampling site in Beijing city from 15 December 2015 to 14 January 2016. The variation trends of NMHCs concentrations in Beijing during the wintertime were basically identical and exhibited significant fluctuation, which were attributed to the variation of the meteorological

conditions. The top ten NMHCs species during the wintertime in Beijing were mainly C2-C5 alkanes, C2-C3 alkenes, acetylene, benzene and toluene. The remarkable difference of the diurnal variations of alkanes, alkenes, aromatics and acetylene between clear days and haze days indicated that the relative contribution of the vehicular emission to atmospheric NMHCs depended on the pollution status. The distinct diurnal variations of the propane/propene ratio indicated that relatively fast consumption of propene by OH radical and $O_3$ in the daytime and by $NO_3$ and $O_3$ in the nighttime. The relatively high concentrations of OH radicals in haze days could accelerate oxidation of gas species and further promoted formation of secondary particles. Both the correlation coefficients of typical hydrocarbons pairs and PMF analysis revealed that coal combustion (probably domestic coal combustion) was an important source for atmospheric NMHCs during wintertime in Beijing, especially in haze days. Therefore, the application of effective control measures for mitigating the serious emissions from prevailingly domestic coal combustion around Beijing in winter are urgent to improve the air quality in Beijing city.

## Author contributions

[⊥] These authors contributed equally.

## Acknowledgements

This work was supported by the National Natural Science Foundation of China (No. 91544211, 21477142, 41575121, 41203070); the Special Fund for Environmental Research in the Public Interest (No. 201509002), and the National Key Research and Development Program of China (2016YFC0202200).

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

**Figures Index and Tables Index**

**Figures:**

Figure 1. Time series of measured NMHCs, PM$_{2.5}$, O$_3$, visibility, relative humidity, temperature and wind speed. The shaded areas indicate pollution episodes: 17-22 December (Cyan), 25-26 December (Yellow), 27-29 December (Green) and 31 December-3 January (LT Gray).

Figure 2. Diurnal variations of alkanes, acetylene, alkenes, aromatics and TNMHC during (A) clear days, (B) light haze days and (C) heavy haze days

Figure 3. Diurnal variations of propane/propene ratios during clear days, light haze days and heavy haze days

Figure 4. Ratios and linear correlation coefficients (R$^2$) between (a) o-xylene and m,p-xylene, (b) cis-2-butene and trans-2-butene, (c) propane and n-butane, and (d) propane and isobutane during clear days (in black), light haze days (in red) and heavy haze days (in blue)

Figure 5. Ratios and linear correlation coefficients (R$^2$) between (a) isopentane and n-pentane, (b) propane and isopentane, (c) benzene and toluene, and (d) benzene and ethylbezene during clear days (in black), light haze days (in red) and heavy haze days (in olive)

Figure 6. Source profiles (percentage of factor total) resolved from PMF in Beijing

Figure 7. Time series of the contributions from gasoline related emissions, diesel exhaust, coal combustion, acetylene-related emission and consumer and household products to atmospheric NMHCs

Figure 8. The diurnal variations of the contributions from the five factors to atmospheric NMHCs (left) and source apportionment of NMHCs (right) in Beijing during clear days, light haze days and heavy haze days

**Tables:**

Table 1. Classification of pollution status and the corresponding meteorological conditions as well as the date

Table 2. The method detection limit (MDL), mean concentrations and standard deviations of NMHCs during clear days, light haze days and heavy haze days (ppbv)

Table 3. Comparisons of the top ten NMHCs in Beijing with other cities in China (ppbv)

Table 4. Emission ratios of NMHCs pairs in Beijing and other regions and comparisons with vehicle emissions

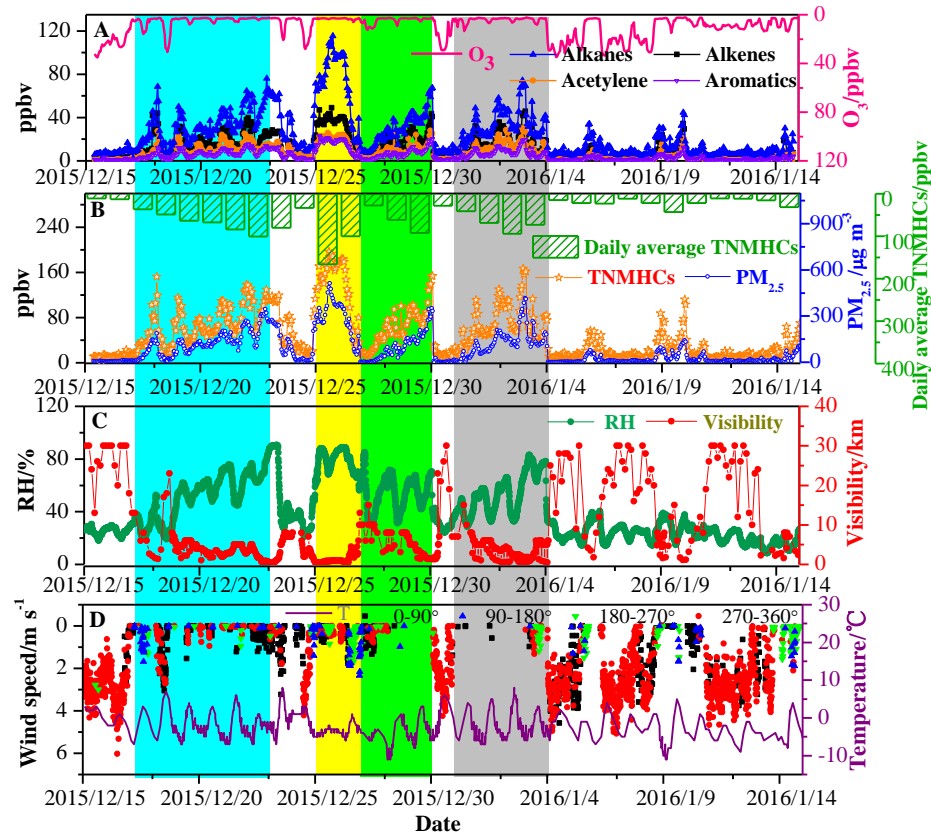

**Fig. 1 Time series of measured NMHCs, PM2.5, O₃, visibility, relative humidity, temperature and wind speed. The shaded areas indicate pollution episodes: 17-22 December (Cyan), 25-26 December (Yellow), 27-29 December (Green) and 31 December-3 January (LT Gray).**

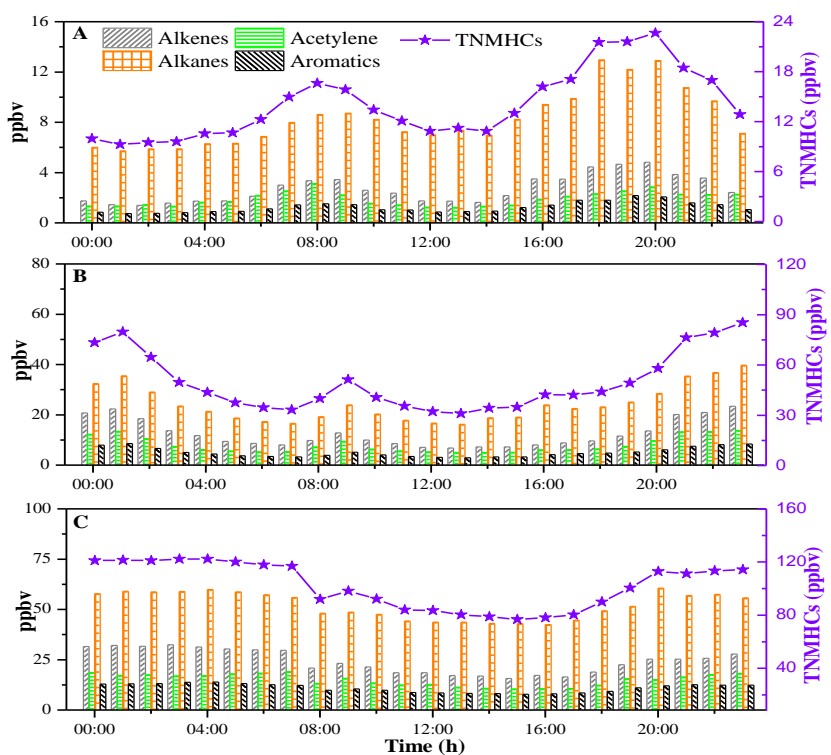

**Fig. 2 Diurnal variations of alkanes, acetylene, alkenes, aromatics and TNMHC during (A) clear days, (B) light haze days and (C) heavy haze days**

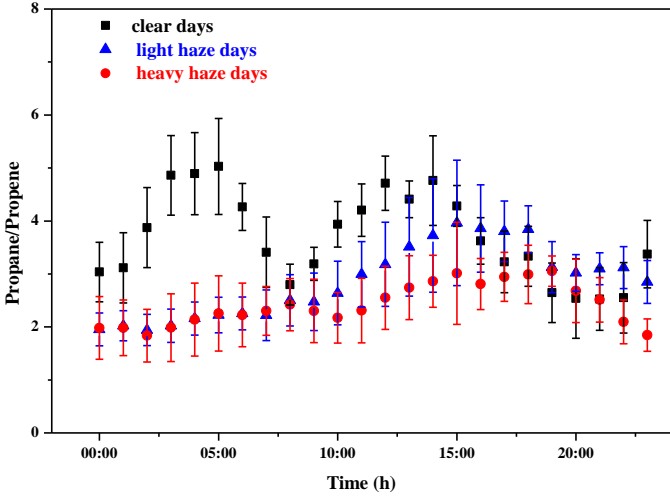

**Fig. 3 Diurnal variations of propane/propene ratios during clear days, light haze days and heavy haze days**

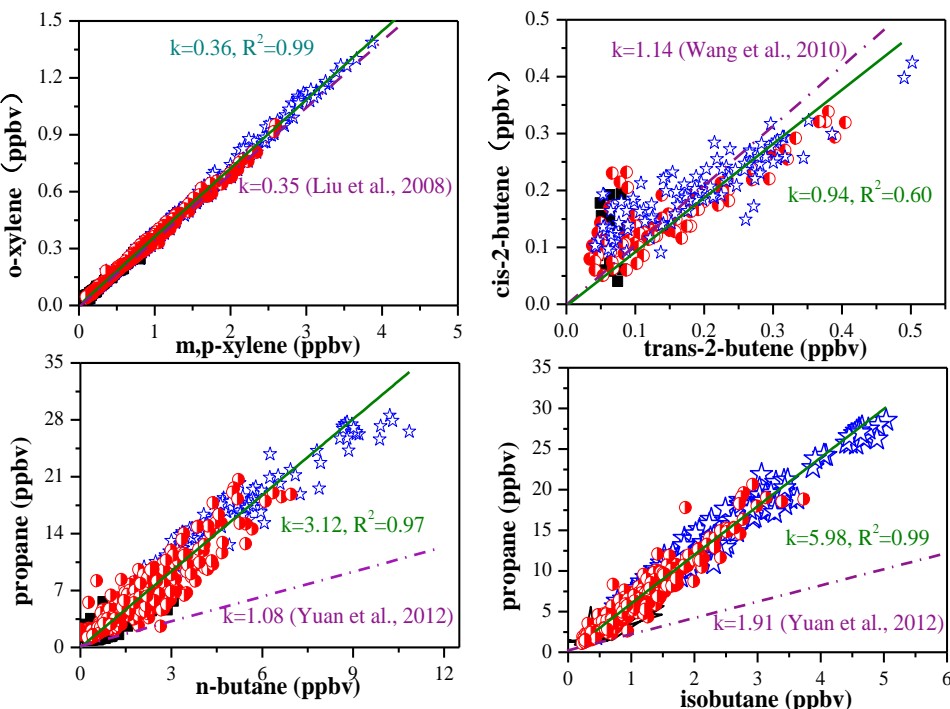

**Fig. 4 Ratios and linear correlation coefficients ($R^2$) between (A) o-xylene and m,p-xylene, (B) cis-2-butene and trans-2-butene, (C) propane and n-butane, and (D) propane and isobutane during clear days (in black), light haze days (in red) and heavy haze days (in blue)**

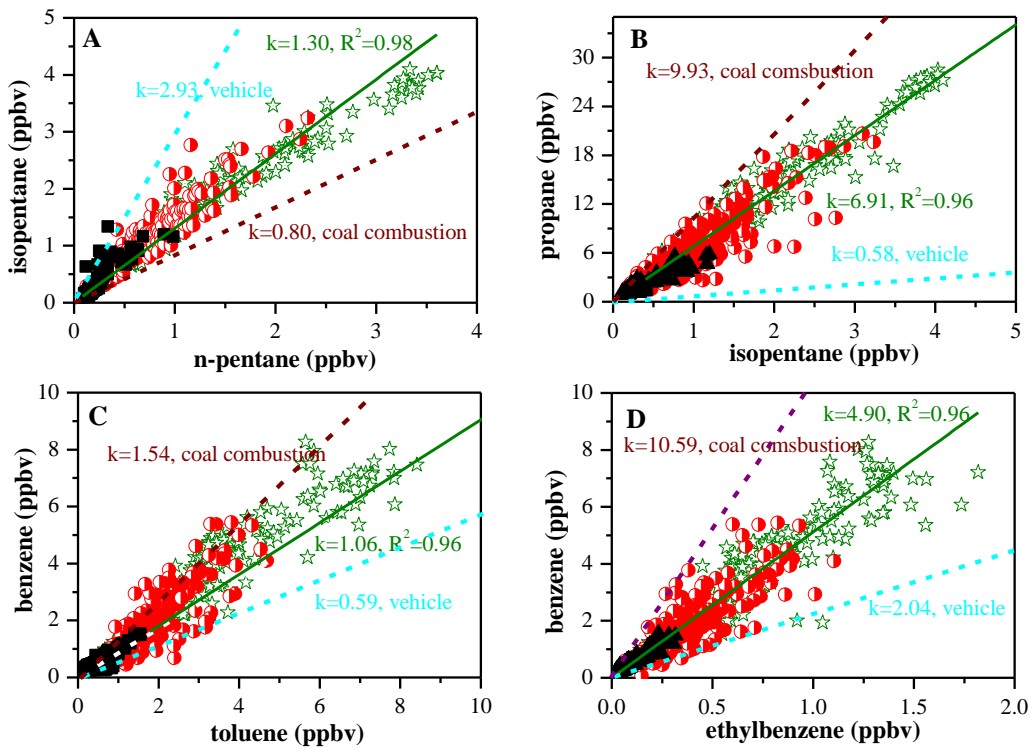

**Fig. 5 Ratios and linear correlation coefficients ($R^2$) between (A) isopentane and n-pentane, (B) propane and isopentane, (C) benzene and toluene, and (D) benzene and ethylbezene during clear days (in black), light haze days (in red) and heavy haze days (in olive)**

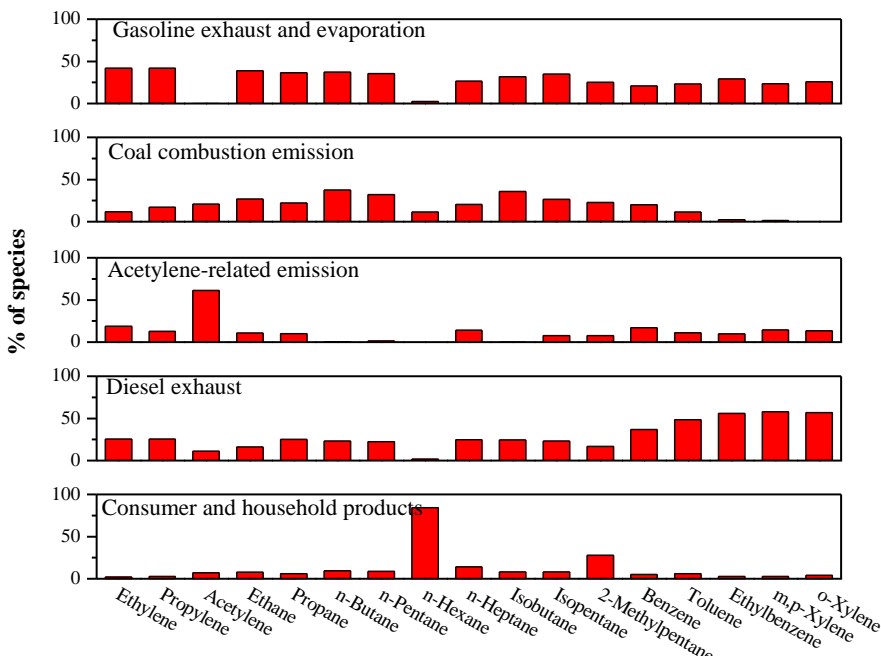

**Fig. 6 Source profiles (percentage of factor total) resolved from PMF in Beijing**

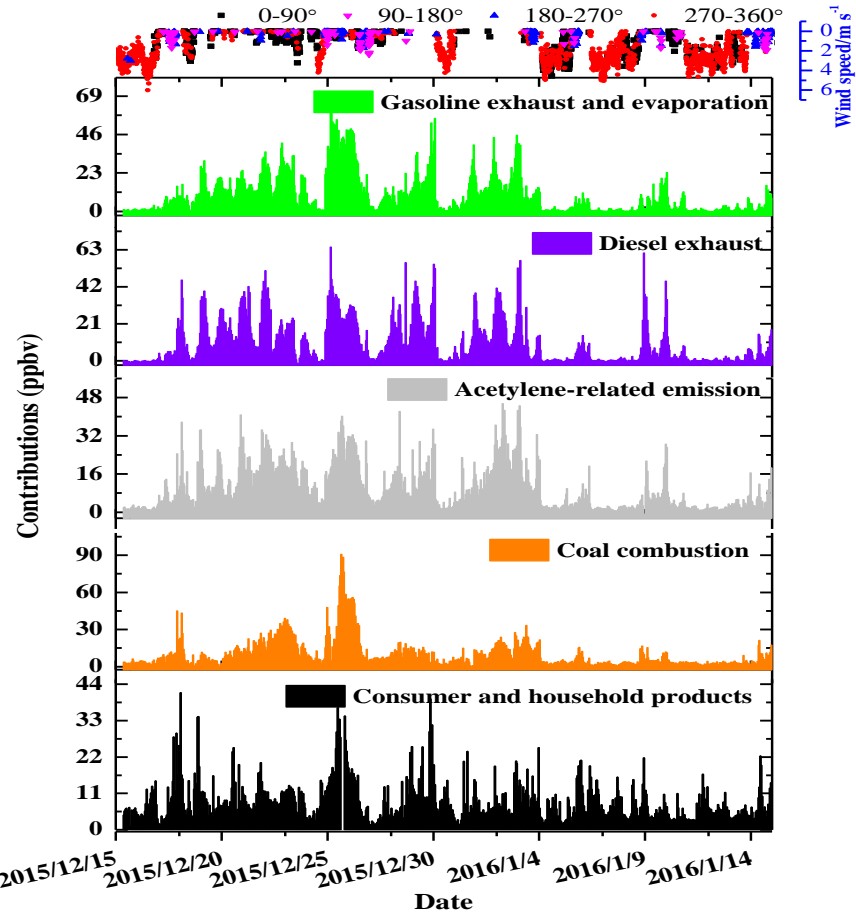

**Fig.7 Time series of the contributions from gasoline related emissions, diesel exhaust, coal combustion, acetylene-related emission and consumer and household products to atmospheric NMHCs**

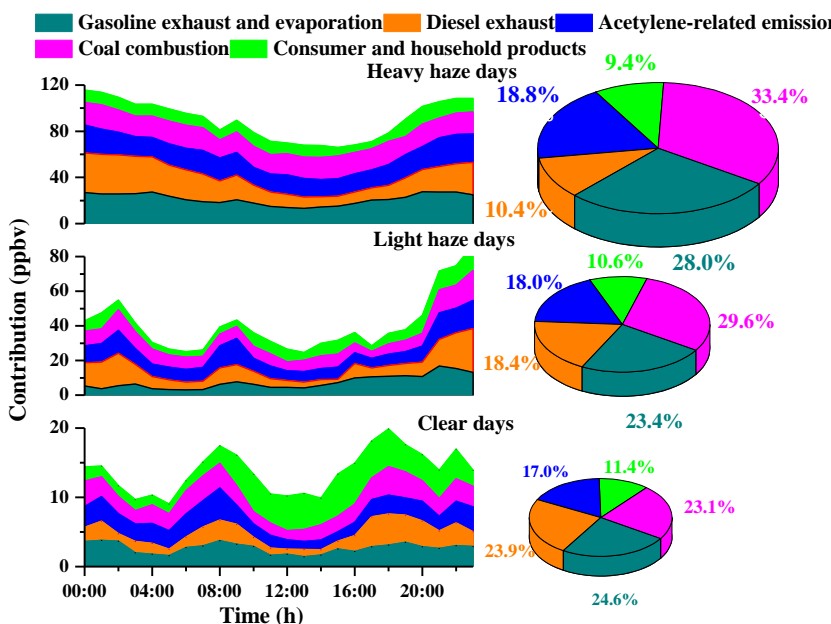

**Fig. 8 The diurnal variations of the contributions from the five factors to atmospheric NMHCs (left) and source apportionment of NMHCs (right) in Beijing during clear days, light haze days and heavy haze days**

**Table 1. Classification of pollution status and the corresponding meteorological conditions as well as the date**

| Pollution status | Visibility /Km | T /℃ | RH /% | Wind speed /m·s$^{-1}$ | Date |
|---|---|---|---|---|---|
| Heavy haze days | 1.41±1.76 | 0.83±2.86 [a] | 59.17±16.19 [a] | 0.19±0.35 [a] | 2015/12/19-23,201512/25-26, 2015/12/29, 2016/01/1-3 |
| | | -0.66±1.90 [b] | 65.94±13.09 [b] | 0.07±0.16 [b] | |
| Light haze days | 6.81±5.37 | -0.72±4.04 [a] | 24.39±11.28 [a] | 0.37±0.68 [a] | 2015/12/17-18, 2015/12/24, 2015/12/27-28, 2015/12/31, 2016/01/9, 2016/01/14 |
| | | -1.19±3.28 [b] | 31.92±14.58 [b] | 0.07±0.23 [b] | |
| Clear days | 19.96±9.7 | 1.63±2.18 [a] | 20.35±6.01 [a] | 2.02±1.29 [a] | 2015/12/15-16, 2015/12/30, 2016/01/4-8, 2016/01/10-13 |
| | | 0.20±2.11 [b] | 26.26±7.56 [b] | 1.78±1.51 [b] | |

[a] daytime; [b] nighttime

**Table 2 The method detection limit (MDL), mean concentrations and standard deviations of NMHCs during clear days, light haze days and heavy haze days (ppbv)**

| Compound | Clear days | Light haze days | Heavy haze days | MDL |
|---|---|---|---|---|
| Ethylene* | 2.43±3.32 | 6.54±5.02 | 15.14±7.01 | 0.08 |
| Propene* | 0.89±1.52 | 2.35±2.31 | 4.51±2.42 | 0.09 |
| 1-Butene | 0.19±0.19 | 0.44±0.29 | 0.78±0.38 | 0.05 |
| Trans-2-Butene | 0.11±0.03 | 0.12±0.04 | 0.15±0.08 | 0.05 |
| Cis-2-Butene | 0.12±0.02 | 0.12±0.05 | 0.17±0.07 | 0.05 |
| 1-penene | 0.07±0.04 | 0.11±0.05 | 0.19±0.09 | 0.05 |
| Isoprene | 0.07±0.04 | 0.12±0.05 | 0.16±0.05 | 0.05 |
| Trans-2-Pentene | 0.06±0.02 | 0.07±0.03 | 0.09±0.04 | 0.05 |
| Cis-2-Pentene | 0.16±0.23 | 0.22±0.21 | 0.51±0.34 | 0.05 |
| 1-Hexene | 0.11±0.01 | 0.11±0.02 | 0.14±0.05 | 0.03 |
| Ethane* | 3.71±2.79 | 8.03±4.66 | 17.63±8.48 | 0.09 |
| Propane* | 2.12±2.02 | 5.18±3.37 | 12.52±6.01 | 0.08 |
| n-Butane* | 0.73±0.81 | 1.76±1.26 | 3.71±2.14 | 0.06 |
| n-Pentane* | 0.26±0.22 | 0.54±0.31 | 1.31±0.78 | 0.05 |
| n-Hexane* | 0.93±0.79 | 0.81±0.86 | 1.04±0.69 | 0.03 |
| n-Heptane* | 0.07±0.06 | 0.15±0.08 | 0.35±0.18 | 0.03 |
| n-Octane | 0.03±0.02 | 0.05±0.03 | 0.09±0.05 | 0.02 |
| Nonane | 0.03±0.02 | 0.05±0.03 | 0.09±0.04 | 0.02 |
| n-Decane | 0.02±0.01 | 0.03±0.02 | 0.06±0.03 | 0.02 |
| n-Undecane | 0.03±0.02 | 0.03±0.01 | 0.04±0.01 | 0.03 |
| Dodecane | 0.05±0.04 | 0.05±0.03 | 0.05±0.03 | 0.09 |
| Isobutane* | 0.44±0.41 | 0.91±0.59 | 2.05±1.04 | 0.07 |
| Isopentane* | 0.39±0.38 | 0.83±0.51 | 1.76±0.88 | 0.05 |

| | | | |
|---|---|---|---|
| 2,2-Dimethylbutane | 0.06±0.04 | 0.08±0.07 | 0.13±0.1 | 0.04 |
| Cyclopentane | 0.07±0.05 | 0.16±0.08 | 0.29±0.13 | 0.05 |
| 2,3-Dimethylbutane | 0.03±0.03 | 0.07±0.05 | 0.09±0.05 | 0.04 |
| 2-Methylpentane* | 0.16±0.14 | 0.36±0.26 | 0.59±0.33 | 0.04 |
| 3-Methylpentane | 0.18±0.13 | 0.24±0.18 | 0.42±0.26 | 0.03 |
| Methylcyclopentane | 0.21±0.15 | 0.32±0.24 | 0.65±0.38 | 0.03 |
| 2,4-Dimethylpentane | 0.03±0.01 | 0.04±0.01 | 0.05±0.02 | 0.03 |
| Cyclohexane | 0.07±0.05 | 0.13±0.13 | 0.24±0.13 | 0.03 |
| 2-Methylhexane | 0.05±0.03 | 0.09±0.05 | 0.18±0.08 | 0.03 |
| 3-Methylhexane | 0.07±0.08 | 0.14±0.09 | 0.36±0.21 | 0.03 |
| 2,2,4-Trimethylpentane | 0.06±0.05 | 0.14±0.07 | 0.22±0.09 | 0.02 |
| Methylcyclohexane | 0.05±0.04 | 0.11±0.07 | 0.26±0.14 | 0.03 |
| 2,3,4-Trimethylpentane | 0.02±0.02 | 0.05±0.03 | 0.07±0.03 | 0.03 |
| 2-Methylheptane | 0.03±0.01 | 0.05±0.03 | 0.08±0.03 | 0.02 |
| 3-Methylheptane | 0.02±0.01 | 0.03±0.02 | 0.05±0.02 | 0.02 |
| Benzene* | 0.59±0.72 | 1.33±0.96 | 3.54±1.76 | 0.03 |
| Toluene* | 0.55±0.66 | 1.34±0.83 | 3.18±1.72 | 0.03 |
| Ethylbenzene* | 0.1±0.14 | 0.27±0.18 | 0.68±0.34 | 0.02 |
| m,p-Xylene* | 0.24±0.35 | 0.66±0.45 | 1.56±0.81 | 0.02 |
| Styrene | 0.06±0.06 | 0.13±0.09 | 0.25±0.14 | 0.03 |
| o-Xylene* | 0.09±0.12 | 0.24±0.16 | 0.57±0.29 | 0.03 |
| Isopropylbenzene | 0.01±0.01 | 0.02±0.01 | 0.02±0.01 | 0.02 |
| n-Propylbenzene | 0.02±0.01 | 0.02±0.01 | 0.04±0.02 | 0.03 |
| m-Ethyltoluene | 0.04±0.03 | 0.08±0.04 | 0.14±0.06 | 0.02 |
| 1,3,5-Thrimethylbenzene | 0.02±0.01 | 0.04±0.02 | 0.06±0.02 | 0.03 |
| o-Ethyltoluene | 0.02±0.01 | 0.03±0.02 | 0.06±0.03 | 0.03 |
| 1,2,4-Thrimethylbenzene | 0.04±0.04 | 0.11±0.06 | 0.19±0.09 | 0.03 |
| 1,2,3-Thrimethylbenzene | 0.02±0.01 | 0.03±0.02 | 0.06±0.04 | 0.03 |
| m-Diethylbenzene | 0.02±0.01 | 0.03±0.01 | 0.03±0.01 | 0.03 |
| p-Diethylbenzene | 0.02±0.02 | 0.02±0.01 | 0.03±0.01 | 0.03 |
| acetylene* | 2.51±2.86 | 6.72±4.71 | 13.69±6.09 | 0.10 |
| alkenes | 3.67±1.78 | 10.01±3.21 | 21.84±6.12 | |
| alkanes | 9.52±2.61 | 20.17±4.90 | 44.83±16.33 | |
| aromatics | 1.58±0.67 | 3.84±1.08 | 9.63±3.28 | |
| TNMHCs | 17.05±5.87 | 40.46±10.92 | 89.98±28.40 | |

* the compounds selected for PMF analysis

**Table 3. Comparisons of the top ten NMHCs in Beijing with other cities in China (ppbv)**

| | This study | | | 43Cities | NJ | GZ | SH | FS[a] | LZ | BJ | BJ[a] |
|---|---|---|---|---|---|---|---|---|---|---|---|
| | The range | AVG | Haze | | | | | | | | |
| ethane | 1.89-44.34 | 9.68 | 13.46 | 3.7-17.0 | 6.90 | 3.66 | - | 18.52 | - | 4.37 | 2.26 |
| ethylene | 0.12-31.65 | 7.91 | 11.37 | 2.1-34.8 | 5.70 | 2.99 | - | 20.58 | - | 2.33 | 6.63 |
| acetylene | 0.40-30.86 | 7.50 | 10.60 | 2.9-58.3 | 3.12 | - | - | 23.38 | - | 2.17 | 5.47 |
| propane | 0.86-28.51 | 6.57 | 9.43 | 1.5-20.8 | 3.30 | 4.34 | 5.16 | 12.98 | 3.40 | 2.44 | 5.45 |
| propene | 0.14-24.10 | 2.55 | 3.54 | 0.2-8.2 | 2.50 | 1.32 | 1.70 | 6.84 | 2.43 | - | 3.32 |
| n-butane | 0.09-14.27 | 2.10 | 2.86 | 0.6-18.8 | 1.70 | 3.07 | 1.69 | 3.76 | 1.75 | 1.43 | 3.49 |
| benzene | 0.07-8.27 | 1.81 | 2.59 | 0.7-10.4 | 3.10 | 0.62 | 2.00 | 4.05 | 1.94 | 0.82 | 2.54 |
| toluene | 0.12-8.41 | 1.67 | 2.38 | 0.4-11.2 | 2.10 | 4.59 | 4.86 | 10.98 | 1.01 | 1.33 | 2.97 |
| isobutane | 0.10-5.03 | 1.13 | 1.55 | 0.4-4.6 | 1.51 | 2.67 | 1.20 | 3.02 | 2.43 | 1.03 | 2.50 |
| isopentane | 0.08-6.03 | 1.00 | 1.36 | 0.3-18.8 | 1.12 | 1.72 | 1.63 | 13.07 | 2.43 | 0.99 | 4.06 |

43 Cities, China, 2001/01-2001/02 (Barletta et al., 2005); NJ, Nanjing, 2011/03-2012/02 (An et al., 2014); GZ, Guangzhou, 2011/06-2012/05 (Zou et al., 2015); SH, Shanghai, 2006/12-2007/02 (Song et al., 2012); FS, Foshan, 2008/12 (Guo et al., 2011b); LZ, Lanzhou, 2013/06-2013/08 (Jia et al., 2016); BJ, Beijing, 2014/05 (Li et al., 2015); BJ, Beijing, 2006/08 (Guo et al., 2012).

[a] haze days.

- data were not available in the relative reference.

**Table 4 Emission ratios of NMHCs pairs in Beijing and other regions and comparisons with vehicle emissions**

| | This study | 43 Chinese cities | Vehicle emissions | Beijing | | Pearl River Delta | | Houston | Mexico City | Northeast US | Tokyo |
|---|---|---|---|---|---|---|---|---|---|---|---|
| | Slope ($R^2$) | Slope [a] | ratio | Slope [b] | Ratio [b] | Slope [c] | Ratio [d] | Ratio [e] | ratio | ratio | ratio |
| benzene/toluene | 1.06 (0.96) | > 1.18  ~ 0.70 | | | 0.43/1.52 [f], 0.38/0.88 | | 0.36 | | | | |
| benzene/acetylene | 0.22 (0.81) | 0.26  0.13 | 0.62 [b] | 0.25/0.27 | 0.27/0.34 | 0.48 | 0.48 | | 0.3 [g] | 0.17/0.30 [g] | 0.29 [g] |
| ethylene/acetylene | 1.08 (0.91) | 1.01  0.76 | | | 0.66/1.00 | | 0.80 | | | | |
| benzene/ethylbenzene | 4.90 (0.91) | 4.91  2.04 | | | 0.93/2.4 | | 1.90 | | | | |
| toluene/ethylene | 0.20 (0.88) | 0.31 | 0.76 [b] | 0.63/0.68 | 0.61/1.26 | 0.46 | 1.67 | | 0.67 [g] | 0.48/0.83 [g] | 1.11 [g] |
| benzene/ethylene | 0.23 (0.96) | 0.17 | | | 0.29/0.47 | | 0.61 | | | | |
| toluene/acetylene | 0.22 (0.81) | 0.24 | | | 0.43/0.83 | | 0.26 | | | | |
| ethylbenzene/toluene | 0.21 (0.97) | | 0.24 [h] | | 0.31/0.37 | 0.20 | 0.19 | 0.14 | 0.12 [i] | | |
| o-xylene/m,p-xylene | 0.36 (0.99) | | 0.35 [h] | | 0.28/0.60 | 0.41 | 0.58 | 0.37 | 0.4 [i] | | |
| propane/toluene | 3.91 (0.92) | | 0.08/0.98 [h] | | 1.13/3.18 | 0.32 | 0.69 | | | | |
| propane/acetylene | 0.93 (0.82) | | 0.06/1.80 [h] | | 0.65/1.51 | 0.42 | 0.92 | | | 2.19 [j] | 2.90 [k] |
| propane/isobutane | 5.98 (0.95) | | 0.74/3.85 [h] | | 1.65/1.94 | 1.91 | 2.00 | | | | |
| propane/n-butane | 3.12 (0.94) | | 0.49/1.91 [h] | | 1.52/1.97 | 1.08 | 1.63 | | | | |
| propane/isopentane | 6.91 (0.85) | | 0.09/0.58 [h] | | 1.29/1.61 | 1.49 | 2.25 | | | | |
| trans-2-butene/cis-2-butene | 1.06 (0.60) | | 1.14 [b] | 1.13/1.23 | 1/1.6 | 1.11 | | 0.88 | 1.28 [i] | | |

[a] Results from 43 Chinese cities in 2001 (Barletta et al., 2005); [b] Results from Beijing in summer of 2008 (Wang et al., 2010); [c] Results from Guangzhou in 2006 and 2008 (Yuan et al., 2012); [d] Results from Pearl river delta in 2008-2009 (Louie et al., 2013); [e] Results from Houston in 2000 (Jobson et al., 2004); [f] Results from Beijing in 2008-2010 (Liu et al., 2009;Zhang et al., 2012a); [g] Emission ratios in Beijing, northeast US, Mexico City, and Tokyo (Parrish et al., 2009); [h] From a tunnel conducted in Guangzhou (Liu et al., 2008); [i] Geometric mean of the ratios in Mexico City in 2003 (Velasco et al., 2007); [j] Results from the Northeast United States in 2004 (Warneke et al., 2007); [k] Ratios in the winter of 2004 in Tokyo (Shirai et al., 2007).