# Peer review of "The levels, variation characteristics and sources of atmospheric nonmethane hydrocarbon compounds during wintertime in Beijing, China"

_Atmospheric Chemistry and Physics, 2016_

## Referee Comment (RC1) · Anonymous Referee #1 · 23 Dec 2016

General comments:

This paper presents measurements of ambient non-methane hydrocarbons (NMHCs) from an urban site in Beijing. Although the observation was conducted only for one month (from 15 December 2015 to 14 January 2016), more than 700 samples were taken and analyzed using a custom-built online gas chromatograph so that good, quasi-continuous time series of NMHCs were obtained. There were some haze periods during the observation, which makes possible to discuss the NMHCs measurements for different pollution conditions. In addition to the characterization of concentrations

and diurnal variations of NMHCs, the authors show the estimation of wintertime OH and NO3 concentrations, discuss the species vs species ratios and the implication to sources, and present the results of source apportionment from the PMF analysis. It is shown that coal combustion is the most important NMHCs source during haze days in winter. The data presented in this paper are of high quality and valuable for atmospheric environmental studies. The results, in particular, the importance of coal combustion to NMHCs and the concentrations of OH and NO3 in winter in Beijing, are not previously reported. In general, the paper is well structured and written. The paper can be improved by appropriately addressing the following major and minor issues. I recommend publication of this paper in ACP after revisions

Specific comments:

(1) Given the inhomogeneous sources distributions and the combination with the winter meteorological conditions, particularly wind direction (e.g., Lin et al., 2011, Zhang et al., 2014), not only wind speed but also wind direction should be used for the interpretation of the NMHCs measurements. Source apportionment suggests that the most important source in haze conditions is coal combustion emission. And it is mentioned that coal combustion is prevailing for heating and cooking by farmers in rural areas. Then the question is: does the dependence of the concentrations of NMHCs and other pollutants on wind direction agrees with the source apportionment and directional distributions of major sources?

(2) The observation period is grouped into clear, light haze and heavy haze days, which are normally closely related with wind speed and direction. Therefore, there might be significant differences in sources impacting the NMHCs at the receptor site. To be able to find the differences, it is suggested to make the PMF analysis separately for the groups of days. However, it seems to me that the PMF analysis was only performed for the entire dataset (section 3.3.2) though the portions of each source are given for different pollution conditions (Fig. 7).

(3) It is suggested to treat the data from 19-22 Dec. 2015 differently since the odd-even plate number rule might have substantially changed the absolute and relative contributions of vehicle emission during this period, which may cause different source apportionment and species ratios.

(4) The interpretation on the different diurnal variations is vague (page 6 lines 20-25). The authors do not show any data of PBL height. The cited references (Quan et al, 2013; Liu et al., 2013) are all about September and the situation may be different in winter months. In addition, the PBL height alone cannot explain the different diurnal patterns. The author states "The boundary layer in clear day is relatively high, which favors for diffusion of pollutants (Gao et al., 2015), and hence, the distinct NMHCs peak values appeared during the two rush hours". If the high PBL in clear day favors the diffusion, emissions from vehicle as well as other sources should be better diluted. Why should the rush hours peaks so protruding? I think the key is the lowest nighttime level of pollution during clear days. It is the lowered nighttime level of pollution that makes the daytime rush hours peaks more evident. If the nighttime PBL were the highest during clear days, the lowest nighttime level of pollution would have been at least partly explained. Unfortunately, the paper presents no PBL height data. However, wind speed data are shown in Table 1 for different pollution conditions. The average wind speed during clear days was nearly twice as high as those during haze days, which could have resulted in the differences. To obtain more robust conclusion, the authors are suggested to calculate the daytime and nighttime wind speed for different pollution conditions. It would be better if they can show data of the PBL height, too.

(5) Species ratios are presented, discussed in terms of emission sources, and used for estimating the OH and NO3 concentrations. While these are good attempts, the authors did not pay attention to uncertainties in use of the ratios. Good correlations can be caused by chemical reactions of NMHCs with OH, NO3 or O3, or simply by atmospheric mixing or dilution (e.g., Parrish et al., 1992; McKeen and Liu, 1993). It seems to me that atmospheric mixing is not considered at all in this paper. The results might have been biased by such omission. I suggest that the authors discuss all the assumptions that are needed to make for the use of this ratio technique and the uncertainties associated with their results.

Minor points:

Page 3 line 11: I think the coordinate is that of RCEES not Beijing city so it should be placed directly after RCEES.

Page 3 lines 24-25: change "ramp" to "ramped"

Page 3 line 30-page 4 line 1: how was the detection limit determined? Either give the determination method here or cite the reference, which in it is described.

Page 4 line 3: a citation for US PMF 5.0 is necessary.

Page 4 lines 23-24: "based on both a good fit to the data and the most reasonable results". Please be more detail about this.

Page 5 line 3 and page 21 Table 1: It is better to change the unit of wind speed to m/s.

Page 6 line 10: delete "of the".

Page 7 line 11: did you measure O3? If yes, the data should be shown in Figure 1.

Page 8 line 2: are the results from daily estimation?

Page 8 line 3: it is meaningless to compare the short-term values for ground level with the global average.

References

Lin, W., Xu, X., Ge, B. and Liu, X.: Gaseous pollutants in Beijing urban area during the heating period 2007–2008: variability, sources, meteorological, and chemical impacts, Atmos. Chem. Phys., 11, 8157–8170, 2011.

McKeen, S.A. and Liu, S.C.: Hydrocarbon ratios and photochemical history of air

masses, Geophys. Res. Lett., 20, 2363–2366, 1993.

Parrish, D.D., Hahn, C.J., Williams, E.J., Norton, R.B., Fehsenfeld, F.C., Singh, H.B., Shetter, J.D., Gandrud, B.W., Ridley, B.A.: Indications of photochemical histories of pacific air masses from measurements of atmospheric trace species at Point Arena, California, J. Geophys. Res., 97, 15883-15902, 1992.

Zhang, H., Xu, X., Lin, W., Wang, Y.: Wintertime peroxyacetyl nitrate (PAN) in the megacity Beijing: Role of photochemical and meteorological processes, J. Environ. Sci., 26, 83–96, 2014.

---

## Referee Comment (RC2) · Anonymous Referee #3 · 12 Jun 2017

This manuscript describes NMHCs measurements at an urban site in Beijing city in winter in order to identify their levels, variation characteristics, and their sources. Therefore, the authors applied a widely-used receptor model, PMF, to investigate sources of NMHCs in winter at this site. The concentrations and their temporal variation were analyzed according to different meteorological situations (clear days and haze days). The overall analysis shows the important contribution of coal combustion emissions to the NMHCs in winter especially in haze days. The paper is well written and organized. I would however, like to see some revisions before it is accepted for publication in ACP.

[Figure]

Major revisions

1-Page 4, line 1: specify how the detection limit is determined.

2-Add a table with all the measured NMHCs (average, median, standard deviation, and detection limit) and indicate which compounds were selected for PMF analysis. The table can be added to supplementary material.

3-Page 4, line 5-6: the authors cited 4 PMF studies in China but there are earlier and more authoritative studies, as the PMF is widely-used and not limited to a specific place. It would be good to integrate some of these as well.

4-I suggest to add wind direction and ozone concentrations in figure 1 and to add analysis accordingly in section 3.1. In addition, pollution roses of some specific compounds can be shown.

5-Page 6, section 3.2.2: Please explain more, in the beginning of the paragraph, why you selected these compounds (propene and propane) because referring to table 4, we can see that the correlation between combustion related species such as benzene/toluene (r2=0.96) and between ethylene/acetylene (r2=0.91) is better than between propane/propene (r2=0.8). Knowing that toluene is more reactive than benzene, and ethylene is more reactive than acetylene.

6-Page 7, line 7: add O2 to the reaction (8) : O3 + NO2 -> NO3 + O2

7-Page 8, section 3.3.1: why did you select the cis-2-butene/trans-2-butene ratio? These species weren't even included in PMF analysis. It seems that these two compounds don't have the same behavior in clear/haze days (figure 4).

8-Page 8, line 10: do you mean gasoline vehicle exhaust or gasoline evaporation related to vehicles?

9-Page 8, the first paragraph (line 10 – 29): It will be better if the ratios from literature are shown in figure 4 as it is the case of figure 5.

10-Page 9, section 3.3.2: the source attribution must be consolidated and more detailed. It will be nice to see diurnal variation of the different factors as well as the time series in different conditions (clear sky, haze days). The PMF results should be also consolidated by using the air quality indicators (like ozone and PM2.5). Adding to that, an analysis of all the factors with wind direction can add more information about the sources and can reveal some point sources such as industries.

11-Page 9, line 15: It is not true that highly reactive NMHCs were excluded because xylenes, ethylene, etc. were included in the PMF analysis so please put other arguments.

12-Page 10, line 1 - 5: how can you explain the correlation of aromatic > C7 with benzene which is a combustion tracer as it correlates also with ethylene and acetylene.

13-Page 10, section 4: please make a brief introduction about the work at the beginning of the paragraph.

Minor revisions:

1-Page 6, line 10 – 12: please rephrase. I think the word "concentration" is lacking.

2-Page 6, line 16-17: rephrase: "…indicated that vehicle exhaust was an important source of NMHCs…"

3-Page 6, line 24: "…which favors accumulation…" remove "the"

4-Keep the same name of compounds in all the manuscript like (propene or propylene; ethane or ethylene…)

5-Page 7, line 27-28: Put this sentence as an explanation before the equations, at line 14.

6-Page 8, line 3: "which is close to…"

7-Page 8, line 23: "… in winter of Beijing are close to those…" and not "closed to"

8-Page 8, line 27: ". . .could also be confirmed. . ." not "been"

9-Page 9, line 15: "with highly reactive", remove "with".

10-Page 9, line 23: "which was in consistent. . ." please clarify, do you mean "inconsistent"?

11-Page 10, line 11: "it is clear. . ." not "clearly"

12-Page 11, line 1: "significant fluctuation. . ." not "significantly"

13-Page 15 and 21, table 1 title: "status" not "statues"

---

## Author Comment (AC1) · 23 Jun 2017

Authors' Response to Referees' Comments

Anonymous Reviewer #1

This paper presents measurements of ambient non-methane hydrocarbons (NMHCs) from an urban site in Beijing. Although the observation was conducted only for one month (from 15 December 2015 to 14 January 2016), more than 700 samples were taken and analyzed using a custom-built online gas chromatograph so that good, quasi-continuous time series of NMHCs were obtained. There were some haze periods during the observation, which makes possible to discuss the NMHCs measurements for different pollution conditions. In addition to the characterization of concentrations and diurnal variations of NMHCs, the authors show the estimation of wintertime OH and NO3 concentrations, discuss the species vs species ratios and the implication to sources, and present the results of source apportionment from the PMF analysis. It is shown that coal combustion is the most important NMHCs source during haze days in winter. The data presented in this paper are of high quality and valuable for atmospheric environmental studies. The results, in particular, the importance of coal combustion to NMHCs and the concentrations of OH and NO3 in winter in Beijing, are not previously reported. In general, the paper is well structured and written. The paper can be improved by appropriately addressing the following major and minor issues. I recommend publication of this paper in ACP after revisions

**Answer:** We appreciate your positive comments about our manuscript. The questions raised by you were responded point by point as followings.

(1) Given the inhomogeneous sources distributions and the combination with the winter meteorological conditions, particularly wind direction (e.g., Lin et al., 2011, Zhang et al., 2014), not only wind speed but also wind direction should be used for the interpretation of the NMHCs measurements. Source apportionment suggests that the most important source in haze conditions is coal combustion emission. And it is mentioned that coal combustion is prevailing for heating and cooking by farmers in rural areas. Then the question is: does the dependence of the concentrations of NMHCs and other pollutants on wind direction agrees with the source apportionment and directional distributions of major sources?

**Answer:** Yes, the concentrations of NMHCs indeed depended on wind direction. The time series of the contributions from the five factors to atmospheric NMHCs were reanalysed (Fig. 7). It is evident that the contribution from coal combustion was the maximum during the most serious pollution episode II (25-26 December 2015) when the wind direction was from southwest, implying that the air parcel transportation from southwest was an important source for NMHCs in Beijing (Wang et al., 2013). According to your valuable suggestion, the corresponding paragraph was revised as following:

In general, the variation trends of the contributions from gasoline related emissions (gasoline exhaust and evaporation), diesel exhaust, coal combustion emissions and acetylene-related emissions to atmospheric NMHCs were closely related with the variation trend of atmospheric

NMHCs measured, while the contribution from the consumer and household products had less correlation with the atmospheric NMHCs measured. The daily emissions from gasoline related sources (gasoline exhaust and evaporation), diesel exhaust, coal combustion sources and acetylene-related sources are usually stable, and hence, the similar variation trends of their contributions to atmospheric NMHCs were mainly ascribed to the variation of meteorological condition. The sources of consumer and household products were suspected to be irregular for explaining the abnormal variation trends of their contributions to atmospheric NMHCs. It should be mentioned that the contribution from coal combustion was the maximum during the most serious pollution episode II (25-26 December 2015) when the wind direction was from southwest, implying that the air parcel transportation from southern was an important source for NMHCs in Beijing (Wang et al., 2013).

[Figure]

Fig.7 The time series of the contributions from gasoline related emissions, diesel exhaust, coal combustion, acetylene-related emission and consumer and household products to atmospheric NMHCs

(2) The observation period is grouped into clear, light haze and heavy haze days, which are normally closely related with wind speed and direction. Therefore, there might be significant differences in sources impacting the NMHCs at the receptor site. To be able to find the differences, it is suggested to make the PMF analysis separately for the groups of days. However, it seems to me that the PMF analysis was only performed for the entire dataset (section 3.3.2) though the portions of each source

are given for different pollution conditions (Fig. 7).

**Answer:** Yes, the sources' contribution to the NMHCs at the receptor site during clear, light haze and heavy haze days were separately derived from the PMF analysis. Because the source profiles for the PMF analysis during different pollution conditions were similar, only the source profiles of the PMF analysis for the entire database are presented in the manuscript. According to your valuable suggestion, the corresponding paragraph was revised as following:

The PMF model was performed based on the 740 samples collected and the NMHCs species with highly reactive or high uncertainty were excluded to reduce the possible bias of the modeling results. Eventually, 17 NMHCs species were selected for the source apportionment analysis since they are the most abundant species and/or are typical tracers of various emission sources. The sources' appointments of atmospheric NMHCs at the receptor site for the clear days, light haze days, heavy haze days and the whole days were separately analysed by the PMF model, and similar source profiles were found. As shown in Fig. 6 for the whole database, five factors were resolved from running the PMF model designated as source 1, source 2, source 3, source 4 and source 5.

(3) It is suggested to treat the data from 19-22 Dec. 2015 differently since the odd even plate number rule might have substantially changed the absolute and relative contributions of vehicle emission during this period, which may cause different source apportionment and species ratios.

**Answer:** According to your valuable suggestion, the PMF analysis was performed separately for the data from 19-22 December 2015 with odd even plate number rule. Compared with the similar pollution days (25-26 December 2015, 29 December 2015, 1-3 January 2016,) but without the odd even plate number rule, the relative contribution of vehicle emission to atmospheric NMHCs during the period of 19-22 December 2015 decreased from 27% to 20%.

(4) The interpretation on the different diurnal variations is vague (page 6 lines 20-25). The authors do not show any data of PBL height. The cited references (Quan et al, 2013; Liu et al., 2013) are all about September and the situation may be different in winter months. In addition, the PBL height alone cannot explain the different diurnal patterns. The author states "The boundary layer in clear day is relatively high, which favors for diffusion of pollutants (Gao et al., 2015), and hence, the distinct NMHCs peak values appeared during the two rush hours". If the high PBL in clear day favors the diffusion, emissions from vehicle as well as other sources should be better diluted. Why should the rush hours peaks so protruding? I think the key is the lowest nighttime level of pollution during clear days. It is the lowered nighttime level of pollution that makes the daytime rush hours peaks more evident. If the nighttime PBL were the highest during clear days, the lowest nighttime level of pollution would have been at least partly explained. Unfortunately, the paper presents no PBL height data. However, wind speed data are shown in Table 1 for different pollution conditions. The average wind speed during clear days was nearly twice as high as those during haze days, which could have resulted in the differences. To obtain more robust conclusion, the authors are suggested to calculate the daytime and nighttime wind speed for different pollution conditions. It would be

better if they can show data of the PBL height, too.

**Answer:** The references about the PBL height under different pollution conditions in winter were cited in the revised manuscript, and the PBL height were indeed the highest during clear days in both daytime and nighttime (Zheng et al., 2015;Lin et al., 2011;Zhang et al., 2014). According to your valuable suggestion, the wind speeds for different pollution days in daytime and nighttime are separately listed in Table 1. Besides the highest PBL height during clear days, the highest wind speed during both nighttime and daytime in clear days also favored diffusion of pollutants, resulting in the lowest levels of atmospheric NMHCs in clear days. Therefore, the statement was rephrased as "Both the relatively high boundary layer and wind speeds could result in the lowest levels of the pollutants during nighttime, which were suspected to make the peak levels of atmospheric NMHCs more evident during daytime rush hours."

Table 1. Classification of pollution statues and the corresponding meteorological conditions as well as the date

| Pollution status | Visibility /Km | T /℃ | RH /% | Wind speed /m·s⁻¹ | Date |
|---|---|---|---|---|---|
| Heavy haze days | 1.41±1.76 | 0.83±2.86 [a]
 -0.66±1.90 [b] | 59.17±16.19 [a]
 65.94±13.09 [b] | 0.19±0.35 [a]
 0.07±0.16 [b] | 2015/12/19-23,201512/25-26,
 2015/12/29, 2016/01/1-3 |
| Light haze days | 6.81±5.37 | -0.72±4.04 [a]
 -1.19±3.28 [b] | 24.39±11.28 [a]
 31.92±14.58 [b] | 0.37±0.68 [a]
 0.07±0.23 [b] | 2015/12/17-18, 2015/12/24,
 2015/12/27-28, 2015/12/31,
 2016/01/9, 2016/01/14 |
| Clear days | 19.96±9.7 | 1.63±2.18 [a]
 0.20±2.11 [b] | 20.35±6.01 [a]
 26.26±7.56 [b] | 2.02±1.29 [a]
 1.78±1.51 [b] | 2015/12/15-16, 2015/12/30,
 2016/01/4-8, 2016/01/10-13 |

[a] daytime; [b] nighttime

(5) Species ratios are presented, discussed in terms of emission sources, and used for estimating the OH and NO3 concentrations. While these are good attempts, the authors did not pay attention to uncertainties in use of the ratios. Good correlations can be caused by chemical reactions of NMHCs with OH, NO3 or O3, or simply by atmospheric mixing or dilution (e.g., Parrish et al., 1992; McKeen and Liu, 1993). It seems to me that atmospheric mixing is not considered at all in this paper. The results might have been biased by such omission. I suggest that the authors discuss all the assumptions that are needed to make for the use of this ratio technique and the uncertainties associated with their results.

**Answer:** According to your valuable suggestion, the uncertainties in use of the Propane/propene ratios were added in Fig.5. The ratios of typical atmospheric pollutants have been widely used as indicators for revealing their sources origination and atmospheric photochemical ageing processes, which could largely counteract the influence of atmospheric dilution (Ho et al., 2004;Barletta et al., 2005;Wang et al., 2010). It should be mentioned that the OH and NO3 derived from the

Propane/propene ratios could only represent their lower limits because of the continue mixing of fresh emissions with the aged air.

[Figure]

Fig. 3 Diurnal variations of propane/propene ratios during clear days, light haze days and heavy haze days

Minor points:

Page 3 line 11: I think the coordinate is that of RCEES not Beijing city so it should be placed directly after RCEES.

**Answer:** Yes! We have corrected the mistake in the revised manuscript.

Page 3 lines 24-25: change "ramp" to "ramped"

**Answer:** Sorry! We have corrected the mistake in the revised manuscript.

Page 3 line 30-page 4 line 1: how was the detection limit determined? Either give the determination method here or cite the reference, which in it is described.

**Answer:** The method detection limits (MDLs) of 0.02-0.10 ppbv for the NMHCs were estimated based on the signal to noise ratio of 3 and enrichment volume of 400 ml. The detail information about the MDLs could be referenced in our previous publication (Liu et al., 2016). The sentence in Page 4, line 1 was revised as following:

The method detection limits (MDLs) of 0.02-0.10 ppbv for the NMHCs were estimated based on the signal to noise ratio of 3 and enrichment volume of 400 ml (Liu et al., 2016a).

Page 4 line 3: a citation for US PMF 5.0 is necessary.

**Answer:** Yes! References were added in the revised manuscript.

Page 4 lines 23-24: "based on both a good fit to the data and the most reasonable results". Please be more detail about this.

**Answer:** Sorry! We have revised the sentence in the revised manuscript:

In this analysis, different numbers of factors were tested to find the optimal fit with the most physically reasonable results. The robust mode was used to reduce the influence of extreme values on the PMF solution.

Page 5 line 3 and page 21 Table 1: It is better to change the unit of wind speed to m/s.

**Answer:** Yes! the unit of wind speed were changed to m/s in the revised manuscript.

Page 6 line 10: delete "of the".

**Answer:** Sorry! We have corrected the mistake in the revised manuscript.

Page 7 line 11: did you measure O3? If yes, the data should be shown in Figure 1.

**Answer:** Yes! ozone concentrations were added in Fig. 1, and the analysis of the pollutants was revised as following:

In contrast to NMHCs and $PM_{2.5}$, ozone concentrations approached to zero during each haze events and reached to the maximum of about 35 ppbv in daytime just after the haze events followed by strong winds from northwest directions (Lin et al., 2011). Although strong winds from northwest directions occurred during the period of 12-14 January 2016, ozone concentrations didn't evidently increase during daytime, implying that ozone formation depended on the pollution levels of its precursors (e.g., NMHCs and NOx).

Page 8 line 2: are the results from daily estimation?

**Answer:** Yes! the daily concentrations of $NO_3$ and OH radicals were estimated.

Page 8 line 3: it is meaningless to compare the short-term values for ground level with the global average.

**Answer:** Sorry! We have corrected the mistake in the revised manuscript.

**References**

Barletta, B., Meinardi, S., Rowland, F. S., Chan, C. Y., Wang, X. M., Zou, S. C., Chan, L. Y., and Blake, D. R.: Volatile organic compounds in 43 Chinese cities, Atmos. Environ.,, 39, 5979-5990, 2005.

Ho, K. F., Lee, S. C., Guo, H., and Tsai, W. Y.: Seasonal and diurnal variations of volatile organic compounds (VOCs) in the atmosphere of Hong Kong, Sci. Total Environ., 322, 155-166, 2004.

Liu, C., Mu, Y., Zhang, C., Zhang, Z., Zhang, Y., Liu, J., Sheng, J., and Quan, J.: Development of gas chromatography-flame ionization detection system with a single column and liquid nitrogen-free for measuring atmospheric C2-C12 hydrocarbons, J. Chromatogr. A, 1427, 134-141, 2016.

Lin, W., Xu, X., Ge, B., and Liu, X.: Gaseous pollutants in Beijing urban area during the heating period 2007–2008: variability, sources, meteorological, and chemical impacts, Atmos. Chem. Phys., 11, 8157-8170, 2011.

Wang, M., Shao, M., Lu, S. H., Yang, Y. D., and Chen, W. T.: Evidence of coal combustion contribution to ambient VOCs during winter in Beijing, Chin. Chem. Lett., 24, 829-832, 2013.

Zhang, H., Xu, X., Lin, W., and Wang, Y.: Wintertime peroxyacetyl nitrate (PAN) in the megacity Beijing: Role of photochemical and meteorological processes, J. Environ. Sci - China, 26, 83-96, 2014.

Zheng, G. J., Duan, F. K., Su, H., Ma, Y. L., Cheng, Y., Zheng, B., Zhang, Q., Huang, T., Kimoto, T., Chang, D., Pöschl, U., Cheng, Y. F., and He, K. B.: Exploring the severe winter haze in Beijing: the impact of synoptic weather, regional transport and heterogeneous reactions, Atmos. Chem. Phys., 15, 2969-2983, 2015.

---

## Author Comment (AC2) · 23 Jun 2017

Authors' Response to Referees' Comments

Anonymous Referee #3

This manuscript describes NMHCs measurements at an urban site in Beijing city in winter in order to identify their levels, variation characteristics, and their sources. Therefore, the authors applied a widely-used receptor model, PMF, to investigate sources of NMHCs in winter at this site. The concentrations and their temporal variation were analyzed according to different meteorological situations (clear days and haze days). The overall analysis shows the important contribution of coal combustion emissions to the NMHCs in winter especially in haze days. The paper is well written and organized. I would however, like to see some revisions before it is accepted for publication in ACP.

**Answer:** We appreciate your positive comments about our manuscript. The questions raised by you were responded point by point as followings.

Major revisions

1-Page 4, line 1: specify how the detection limit is determined.

**Answer:** The method detection limits (MDLs) of 0.02-0.10 ppbv for the NMHCs were estimated based on the signal to noise ratio of 3 and enrichment volume of 400 ml. The detail information about the MDLs could be referenced in our previous publication (Liu et al., 2016). The sentence in Page 4, line 1 was revised as following:

The method detection limits (MDLs) of 0.02-0.10 ppbv for the NMHCs were estimated based on the signal to noise ratio of 3 and enrichment volume of 400 ml (Liu et al., 2016a).

2-Add a table with all the measured NMHCs (average, median, standard deviation, and detection limit) and indicate which compounds were selected for PMF analysis. The table can be added to supplementary material.

**Answer:** According to your valuable suggestion, the average, standard deviation, and detection limit of the measured NMHCs are listed in Table 2, and the compounds selected for PMF analysis were marked as *.

Table 2 The method detection limit (MDL), mean concentrations and standard deviations of NMHCs during clear days, light haze days and heavy haze days (ppbv)

| Compund | Clear days | Light haze days | Heavy haze days | MDL |
|---|---|---|---|---|
| Ethylene* | 2.43±3.32 | 6.54±5.02 | 15.14±7.01 | 0.08 |
| Propene* | 0.89±1.52 | 2.35±2.31 | 4.51±2.42 | 0.09 |
| 1-Butene | 0.19±0.19 | 0.44±0.29 | 0.78±0.38 | 0.05 |
| Trans-2-Butene | 0.11±0.03 | 0.12±0.04 | 0.15±0.08 | 0.05 |
| Cis-2-Butene | 0.12±0.02 | 0.12±0.05 | 0.17±0.07 | 0.05 |

| | | | |
|---|---|---|---|
| 1-penene | 0.07±0.04 | 0.11±0.05 | 0.19±0.09 | 0.05 |
| Isoprene | 0.07±0.04 | 0.12±0.05 | 0.16±0.05 | 0.05 |
| Trans-2-Pentene | 0.06±0.02 | 0.07±0.03 | 0.09±0.04 | 0.05 |
| Cis-2-Pentene | 0.16±0.23 | 0.22±0.21 | 0.51±0.34 | 0.05 |
| 1-Hexene | 0.11±0.01 | 0.11±0.02 | 0.14±0.05 | 0.03 |
| Ethane* | 3.71±2.79 | 8.03±4.66 | 17.63±8.48 | 0.09 |
| Propane* | 2.12±2.02 | 5.18±3.37 | 12.52±6.01 | 0.08 |
| n-Butane* | 0.73±0.81 | 1.76±1.26 | 3.71±2.14 | 0.06 |
| n-Pentane* | 0.26±0.22 | 0.54±0.31 | 1.31±0.78 | 0.05 |
| n-Hexane* | 0.93±0.79 | 0.81±0.86 | 1.04±0.69 | 0.03 |
| n-Heptane* | 0.07±0.06 | 0.15±0.08 | 0.35±0.18 | 0.03 |
| n-Octane | 0.03±0.02 | 0.05±0.03 | 0.09±0.05 | 0.02 |
| Nonane | 0.03±0.02 | 0.05±0.03 | 0.09±0.04 | 0.02 |
| n-Decane | 0.02±0.01 | 0.03±0.02 | 0.06±0.03 | 0.02 |
| n-Undecane | 0.03±0.02 | 0.03±0.01 | 0.04±0.01 | 0.03 |
| Dodecane | 0.05±0.04 | 0.05±0.03 | 0.05±0.03 | 0.09 |
| Isobutane* | 0.44±0.41 | 0.91±0.59 | 2.05±1.04 | 0.07 |
| Isopentane* | 0.39±0.38 | 0.83±0.51 | 1.76±0.88 | 0.05 |
| 2,2-Dimethylbutane | 0.06±0.04 | 0.08±0.07 | 0.13±0.1 | 0.04 |
| Cyclopentane | 0.07±0.05 | 0.16±0.08 | 0.29±0.13 | 0.05 |
| 2,3-Dimethylbutane | 0.03±0.03 | 0.07±0.05 | 0.09±0.05 | 0.04 |
| 2-Methylpentane* | 0.16±0.14 | 0.36±0.26 | 0.59±0.33 | 0.04 |
| 3-Methylpentane | 0.18±0.13 | 0.24±0.18 | 0.42±0.26 | 0.03 |
| Methylcyclopentane | 0.21±0.15 | 0.32±0.24 | 0.65±0.38 | 0.03 |
| 2,4-Dimethylpentane | 0.03±0.01 | 0.04±0.01 | 0.05±0.02 | 0.03 |
| Cyclohexane | 0.07±0.05 | 0.13±0.13 | 0.24±0.13 | 0.03 |
| 2-Methylhexane | 0.05±0.03 | 0.09±0.05 | 0.18±0.08 | 0.03 |
| 3-Methylhexane | 0.07±0.08 | 0.14±0.09 | 0.36±0.21 | 0.03 |
| 2,2,4-Trimethylpentane | 0.06±0.05 | 0.14±0.07 | 0.22±0.09 | 0.02 |
| Methylcyclohexane | 0.05±0.04 | 0.11±0.07 | 0.26±0.14 | 0.03 |
| 2,3,4-Trimethylpentane | 0.02±0.02 | 0.05±0.03 | 0.07±0.03 | 0.03 |
| 2-Methylheptane | 0.03±0.01 | 0.05±0.03 | 0.08±0.03 | 0.02 |
| 3-Methylheptane | 0.02±0.01 | 0.03±0.02 | 0.05±0.02 | 0.02 |
| Benzene* | 0.59±0.72 | 1.33±0.96 | 3.54±1.76 | 0.03 |
| Toluene* | 0.55±0.66 | 1.34±0.83 | 3.18±1.72 | 0.03 |

| | | | | |
|---|---|---|---|---|
| Ethylbenzene* | 0.1±0.14 | 0.27±0.18 | 0.68±0.34 | 0.02 |
| m,p-Xylene* | 0.24±0.35 | 0.66±0.45 | 1.56±0.81 | 0.02 |
| Styrene | 0.06±0.06 | 0.13±0.09 | 0.25±0.14 | 0.03 |
| o-Xylene* | 0.09±0.12 | 0.24±0.16 | 0.57±0.29 | 0.03 |
| Isopropylbenzene | 0.01±0.01 | 0.02±0.01 | 0.02±0.01 | 0.02 |
| n-Propylbenzene | 0.02±0.01 | 0.02±0.01 | 0.04±0.02 | 0.03 |
| m-Ethyltoluene | 0.04±0.03 | 0.08±0.04 | 0.14±0.06 | 0.02 |
| 1,3,5-Thrimethylbenzene | 0.02±0.01 | 0.04±0.02 | 0.06±0.02 | 0.03 |
| o-Ethyltoluene | 0.02±0.01 | 0.03±0.02 | 0.06±0.03 | 0.03 |
| 1,2,4-Thrimethylbenzene | 0.04±0.04 | 0.11±0.06 | 0.19±0.09 | 0.03 |
| 1,2,3-Thrimethylbenzene | 0.02±0.01 | 0.03±0.02 | 0.06±0.04 | 0.03 |
| m-Diethylbenzene | 0.02±0.01 | 0.03±0.01 | 0.03±0.01 | 0.03 |
| p-Diethylbenzene | 0.02±0.02 | 0.02±0.01 | 0.03±0.01 | 0.03 |
| acetylene* | 2.51±2.86 | 6.72±4.71 | 13.69±6.09 | 0.10 |
| alkenes | 3.67±1.78 | 10.01±3.21 | 21.84±6.12 | |
| alkanes | 9.52±2.61 | 20.17±4.90 | 44.83±16.33 | |
| aromatics | 1.58±0.67 | 3.84±1.08 | 9.63±3.28 | |
| TNMHCs | 17.05±5.87 | 40.46±10.92 | 89.98±28.40 | |

* the compounds selected for PMF analysis

3-Page 4, line 5-6: the authors cited 4 PMF studies in China but there are earlier and more authoritative studies, as the PMF is widely-used and not limited to a specific place. It would be good to integrate some of these as well.

**Answer:** According to your valuable suggesting, the earlier and more authoritative references about application of the PMF analysis were cited as following:

The US PMF 5.0 was applied to identify major emission sources of NMHCs sources (Sowlat et al., 2016). PMF is a multivariate factor analysis tool that decomposes a matrix of speciated sample data into two matrices-factor contributions and factor profiles which can be interpreted by an analyst as to what sources are represented based on observations at the receptor site (Guo et al., 2010; Ling et al., 2014; Ou et al., 2015; Shao et al., 2016; Shi et al., 2009;Xie et al., 2008;Lanz et al., 2007;Zhang et al., 2013).

4-I suggest to add wind direction and ozone concentrations in figure 1 and to add analysis accordingly in section 3.1. In addition, pollution roses of some specific compounds can be shown.

**Answer:** According to your valuable suggesting, wind direction and ozone concentrations were

added in Fig. 1, and the analysis of the pollutants was revised as following:

In contrast to NMHCs and PM$_{2.5}$, ozone concentrations approached to zero during each haze events and reached to the maximum of about 35 ppbv in daytime just after the haze events followed by strong winds from northwest directions. Although strong winds from northwest directions occurred during the period of 12-14 January 2016, ozone concentrations didn't evidently increase during daytime, implying that ozone formation depended on the pollution levels of its precursors (NMHCs and NOx).

[Figure]

Fig. 1 Time series of measured NMHCs, PM$_{2.5}$, O$_3$, visibility, relative humidity, temperature and wind speed. The shaded areas indicate pollution episodes: 17-22 December (Cyan), 25-26 December (Yellow), 27-29 December (Green) and 31 December-3 January (LT Gray).

5-Page 6, section 3.2.2: Please explain more, in the beginning of the paragraph, why you selected these compounds (propene and propane) because referring to table 4, we can see that the correlation between combustion related specsies such as benzene/toluene (r2=0.96) and between ethylene/acetylene (r2=0.91) is better than between propane/propene (r2=0.8). Knowing that toluene is more reactive than benzene, and ethylene is more reactive than acetylene.

**Answer:** Considering the large difference of the reactivity between propane and propene towards NO$_3$ and OH radicals, the diurnal variations of propane/propene ratios were analyzed for revealing the nighttime and daytime reaction processes. Although toluene is more reactive than benzene, and ethylene is more reactive than acetylene towards OH radicals, the small difference of the reaction

reactivity towards $NO_3$ between the two hydrocarbon pairs ($k_{(toluene+NO3)}/k_{(benzene+NO3)}$ = 3.1; $k_{(ethylene+NO3)}/k_{(acetylene+NO3)}$ =2.1) could not reflect the nighttime $NO_3$ chemistry. In addition, the difference of the reaction reactivity towards OH radicals between propane and propene ($k_{(propene+OH)}/k_{(propane+OH)}$ =23.58) is remarkably greater than between the hydrocarbon pairs of benzene/toluene ($k_{(toluene+OH)}/k_{(benzene+OH)}$ =4.61) and ethylene/acetylene ($k_{(ethylene+OH)}/k_{(acetylene+OH)}$ =9.36). According to your valuable suggestion, the sentence of "Considering the large difference of the reactivity between propane and propene towards $NO_3$ and OH radicals, the diurnal variations of propane/propene ratios were analyzed for revealing the nighttime and daytime reaction processes." was added in the beginning of the paragraph.

6-Page 7, line 7: add O2 to the reaction (8) : O3 + NO2 -> NO3 + O2
**Answer:** According to your valuable suggesting, $O_2$ were added in the reaction (8).

7-Page 8, section 3.3.1: why did you select the cis-2-butene/trans-2-butene ratio? These species weren't even included in PMF analysis. It seems that these two compounds don't have the same behavior in clear/haze days (figure 4).
**Answer:** Previous studies have used the cis-2-butene/trans-2-butene ratio as the indicator for gasoline vehicle emissions, and hence the ratio was also analyzed in this study for comparison. Because the levels of cis-2-butene and trans-2-butene during clear days were close to the detection limits (0.05 ppbv) of the instrument, the large uncertainties of the data measured in clear days might be the reason for the different behaviors of the two compounds between clear days and haze days. Therefore, the species were not included in PMF analysis.

8-Page 8, line 10: do you mean gasoline vehicle exhaust or gasoline evaporation related to vehicles?
**Answer:** Sorry for the unclear description, and the sentence was revised as following:
The ratios of o-xylene/m,p-xylene and cis-2-butene/trans-2-butene have been widely used as the indicators for gasoline vehicle exhaust emissions.

9-Page 8, the first paragraph (line 10 – 29): It will be better if the ratios from literature are shown in figure 4 as it is the case of figure 5.
**Answer:** According to your valuable suggesting, the slopes from literature are shown in Fig. 4.

[Figure]

Fig. 4 Ratios and linear correlation coefficients ($R^2$) between (A) o-xylene and m,p-xylene, (B) cis-2-butene and trans-2-butene, (C) propane and n-butane, and (D) propane and isobutane during clear days (in black), light haze days (in red) and heavy haze days (in blue)

10-Page 9, section 3.3.2: the source attribution must be consolidated and more detailed. It will be nice to see diurnal variation of the different factors as well as the time series in different conditions (clear sky, haze days). The PMF results should be also consolidated by using the air quality indicators (like ozone and PM2.5). Adding to that, an analysis of all the factors with wind direction can add more information about the sources and can reveal some point sources such as industries.

**Answer:** According to your valuable suggestions, the diurnal variation of the different factors as well as the time series in different conditions were analyzed in the revised manuscript as followings: The time series of the contributions from the five factors to atmospheric NMHCs are shown in Fig. 7. In general, the variation trends of the contributions from gasoline related emissions (gasoline exhaust and evaporation), diesel exhaust, coal combustion emissions and acetylene-related emissions to atmospheric NMHCs were closely related with the variation trend of atmospheric NMHCs measured, while the contribution from the consumer and household products had less correlation with the atmospheric NMHCs measured. The daily emissions from gasoline related sources (gasoline exhaust and evaporation), diesel exhaust, coal combustion sources and acetylene-related sources are usually stable, and hence, the similar variation trends of their contributions to atmospheric NMHCs were mainly ascribed to the variation of meteorological condition. The sources of consumer and household products were suspected to be irregular for explaining the abnormal variation trends of their contributions to atmospheric NMHCs. It should be mentioned that the contribution from coal combustion was the maximum during the most serious pollution episode II (25-26 December 2015) when the wind direction was from southwest, implying that the air parcel

transportation from southern was an important source for NMHCs in Beijing (Wang et al., 2013). The diurnal variations of the contributions from the five factors to atmospheric NMHCs are shown in Fig. 8. Compared with the sources of coal combustion, acetylene-related emissions and consumer and household products, the contributions of the vehicle emissions (gasoline and diesel exhaust) to atmospheric NMHCs during the morning and evening rush hours indeed evidently increased during the clear days and light haze days, but slightly decreased in the morning rush hours during the heavy haze days. The remarkably higher contributions of diesel exhaust than gasoline emissions during the midnight for haze days well reflected the traffic situation, namely, heavy diesel vehicles being only permitted on the road during the midnight in Beijing. The relatively high contributions of consumer and household products to atmospheric NMHCs mainly occurred in clear days during daytime when temperature was relatively high. No distinct diurnal variations of the contributions from coal combustion and acetylene-related emissions to atmospheric NMHCs were found.

[Figure]

Fig.7 The time series of the contributions from gasoline related emissions, diesel exhaust, coal combustion, acetylene-related emission and consumer and household products to atmospheric NMHCs

[Figure]

**Fig. 8** The diurnal variations of the contributions from the five factors to atmospheric NMHCs (left) and source apportionment of NMHCs (right) in Beijing during clear days, light haze days and heavy haze days

11-Page 9, line 15: It is not true that highly reactive NMHCs were excluded because xylenes, ethylene, etc. were included in the PMF analysis so please put other arguments.

**Answer:** Yes, you are right. According to your valuable comment, the sentence was revised as following:

The PMF model was performed based on the data of 740 air samples and the NMHCs species with high uncertainties of the measurement were excluded to reduce the possible bias of the modeling results.

12-Page 10, line 1 - 5: how can you explain the correlation of aromatic > C7 with benzene which is a combustion tracer as it correlates also with ethylene and acetylene.

**Answer:** Yes, your suspicion is logical. The sentence was revised as following:

It is known that these species can be emitted from coal combustion, vehicular exhaust or associated with the solvent emissions of paints, inks, sealant, varnish and thinner for architecture and decoration (Borbon et al., 2002; Guo et al., 2011a). Coal combustion and gasoline exhaust could be excluded as the main contributors to source 4, because aromatics emissions from the two sources are usually accompanied by high emissions of various species with carbon numbers less than six. Solvent emissions could also be excluded due to the relatively high contribution of small molecules such as ethylene and propene in source 4. Based on the PMF analysis for the diurnal variation characters, source 4 is finally attributed to diesel exhaust.

13-Page 10, section 4: please make a brief introduction about the work at the beginning of the

paragraph.

**Answer:** According to your valuable suggestion, a brief introduction was added at the beginning of the paragraph: Atmospheric non-methane hydrocarbon compounds (NMHCs) were measured at a sampling site in Beijing city from 15 December 2015 to 14 January 2016.

Minor revisions:
1- Page 6, line 10 – 12: please rephrase. I think the word "concentration" is lacking.
**Answer:** Sorry! We have corrected the mistake in the revised manuscript.

2- Page 6, line 16-17: rephrase: ". . .indicated that vehicle exhaust was an important source of NMHCs. . ."
**Answer:** Sorry! We have corrected the mistake in the revised manuscript.

3- Page 6, line 24: ". . .which favors accumulation. . ." remove "the"
**Answer:** Sorry! We have corrected the mistake in the revised manuscript.

4-Keep the same name of compounds in all the manuscript like (propene or propylene; ethane or ethylene. . .)
**Answer:** Sorry! We have corrected the mistake in the revised manuscript.

5-Page 7, line 27-28: Put this sentence as an explanation before the equations, at line 14.
**Answer:** Yes! We have put the sentence before the equations.

6-Page 8, line 3: "which is close to. . ."
**Answer:** Sorry! We have corrected the mistake in the revised manuscript.

7-Page 8, line 23: ". . . in winter of Beijing are close to those. . ." and not "closed to"
**Answer:** Sorry! We have corrected the mistake in the revised manuscript.

8-Page 8, line 27: ". . .could also be confirmed. . ." not "been"
**Answer:** Sorry! We have corrected the mistake in the revised manuscript.

9-Page 9, line 15: "with highly reactive", remove "with".
**Answer:** Sorry! We have corrected the mistake in the revised manuscript.

10-Page 9, line 23: "which was in consistent. . ." please clarify, do you mean "inconsistent"?
**Answer:** Sorry! We have corrected the mistake in the revised manuscript.

11-Page 10, line 11: "it is clear. . ." not "clearly"

**Answer:** Sorry! We have corrected the mistake in the revised manuscript.

12-Page 11, line 1: "significant fluctuation. . ." not "significantly"

**Answer:** Sorry! We have corrected the mistake in the revised manuscript.

13-Page 15 and 21, table 1 title: "status" not "statues"

**Answer:** Sorry! We have corrected the mistake in the revised manuscript.

**References**

Borbon, A., Locoge, N., Veillerot, M., Galloo, J., and Guillermo, R.: Characterisation of NMHCs in a French urban atmosphere: overview of the main sources, Sci. Total Environ., 292, 177-191, 2002.

Guo, S., Hu, M., Wang, Z., Slanina, J., and Zhao, Y.: Size-resolved aerosol water-soluble ionic compositions in the summer of Beijing: implication of regional secondary formation, Atmos. Chem. Phys., 10, 947-959, 2010.

Guo, H., Cheng, H., Ling, Z., Louie, P., and Ayoko, G.: Which emission sources are responsible for the volatile organic compounds in the atmosphere of Pearl River Delta?, J. Hazard. Mater., 188, 116-124, 2011a.

Lanz, V. A., Alfarra, M. R., Baltensperger, U., Buchmann, B., Hueglin, C., and Prévôt, A. S. H.: Source apportionment of submicron organic aerosols at an urban site by factor analytical modelling of aerosol mass spectra, Atmos. Chem. Phys., 7, 1503-1522, 2007.

Ling, Z., and Guo, H.: Contribution of VOC sources to photochemical ozone formation and its control policy implication in Hong Kong, Environ. Sci. Policy, 38, 180-191, 2014.

Liu, C., Mu, Y., Zhang, C., Zhang, Z., Zhang, Y., Liu, J., Sheng, J., and Quan, J.: Development of gas chromatography-flame ionization detection system with a single column and liquid nitrogen-free for measuring atmospheric C2-C12 hydrocarbons, J. Chromatogr. A, 1427, 134-141, 2016a.

Liu, Y., Shao, M., Fu, L. L., Lu, S. H., Zeng, L. M., and Tang, D. G.: Source profiles of volatile organic compounds (VOCs) measured in China: Part I, Atmos. Environ., 42, 6247-6260, 2008.

Ou, J., Guo, H., Zheng, J., Cheung, K., Louie, P. K., Ling, Z., and Wang, D.: Concentrations and sources of non-methane hydrocarbons (NMHCs) from 2005 to 2013 in Hong Kong: A multi-year real-time data analysis, Atmos. Environ., 103, 196-206, 2015.

Shi, G.L., Li, X., Feng, Y.C., Wang, Y.Q., Wu, J.H., Li, J., and Zhu, T.: Combined source apportionment, using positive matrix factorization–chemical mass balance and principal component analysis/multiple linear regression–chemical mass balance models, Atmos. Environ., 43, 2929-2937, 2009.

Sowlat, M. H., Hasheminassab, S., and Sioutas, C.: Source apportionment of ambient particle

number concentrations in central Los Angeles using positive matrix factorization (PMF), Atmos. Chem. Phys., 16, 4849-4866, 2016.

Wang, B., Shao, M., Lu, S., Yuan, B., Zhao, Y., Wang, M., Zhang, S., and Wu, D.: Variation of ambient non-methane hydrocarbons in Beijing city in summer 2008, Atmos. Chem. Phys., 10, 5911-5923, 2010.

Wang, M., Shao, M., Lu, S. H., Yang, Y. D., and Chen, W. T.: Evidence of coal combustion contribution to ambient VOCs during winter in Beijing, Chin. Chem. Lett., 24, 829-832, 2013.

Xie, S. D., Liu, Z., Chen, T., and Hua, L.: Spatiotemporal variations of ambient $PM_{10}$ source contributions in Beijing in 2004 using positive matrix factorization, Atmos. Chem. Phys., 8, 2701-2716, 2008.

Yuan, B., Chen, W., Shao, M., Wang, M., Lu, S., Wang, B., Liu, Y., Chang, C., and Wang, B.: Measurements of ambient hydrocarbons and carbonyls in the Pearl River Delta (PRD), China, Atmos. Res., 116, 93-104, 2012.

Zhang, R., Jing, J., Tao, J., Hsu, S. C., Wang, G., Cao, J., Lee, C. S. L., Zhu, L., Chen, Z., Zhao, Y., and Shen, Z.: Chemical characterization and source apportionment of $PM_{2.5}$ in Beijing: seasonal perspective, Atmos. Chem. Phys., 13, 7053-7074, 2013.